# LM-MIXUP: TEXT DATA AUGMENTATION VIA LANGUAGE MODEL BASED MIXUP

## ABSTRACT

Instruction tuning is crucial for aligning Large Language Models (LLMs), yet the quality of instruction-following data varies significantly. While high-quality data is paramount, it is often scarce; conversely, abundant low-quality data is frequently discarded, leading to substantial information loss. Existing data augmentation methods struggle to augment this low-quality data effectively, and the evaluation of such techniques remains poorly defined. To address this, we formally define the task of *Instruction Distillation*: distilling multiple low-quality and redundant inputs into high-quality and coherent instruction-output pairs. Specifically, we introduce a comprehensive data construction pipeline to create MIXTURE, a 144K-sample dataset pairing low-quality or semantically redundant imperfect instruction clusters with their high-quality distillations. We then introduce *LM-Mixup*, by first performing supervised fine-tuning on MIXTURE and then optimizing it with reinforcement learning. This process uses three complementary reward signals: quality, semantic alignment, and format compliance, via Group Relative Policy Optimization (GRPO). We demonstrate that *LM-Mixup* effectively augments imperfect datasets: fine-tuning LLMs on its distilled data, which accounts for only about 3% of the entire dataset, not only surpasses full-dataset training but also competes with state-of-the-art high-quality data selection methods across multiple benchmarks. Our work establishes that low-quality data is a valuable resource when properly distilled and augmented with *LM-Mixup*, significantly enhancing the efficiency and performance of instruction-tuned LLMs.

## 1 INTRODUCTION

In recent years, large language models (LLMs) have achieved notable progress in natural language processing and multimodal understanding (Team et al., 2023; Singhal et al., 2023; Deng et al., 2025; Li et al., 2024b; 2025a). This progress stems not only from improved architectures and larger scales but also from more efficient ways for models to learn and apply knowledge (Patil & Jadon, 2025; Dredze, 2025). While the conventional view holds that high-quality human alignment requires massive annotated data (Kim et al., 2024; Köpf et al., 2023), recent studies show that LLMs acquire most knowledge during pre-training (Brown et al., 2020; Roberts et al., 2020). Only a small, carefully curated dataset is sufficient for effective alignment in instruction tuning or supervised fine-tuning (SFT) (He et al., 2024; Wei et al., 2023), so many works now focus on selecting high-quality data, demonstrating that fine-tuning on such subsets alone can already yield strong performance (Pang et al., 2024; Fu et al., 2025; Jha et al., 2023). This shifts the research focus from "more data" to "better data", emphasizing efficient high-quality data selection for model improvement.

However, high-quality samples are scarce and costly, while real-world datasets contain abundant redundant or low-quality data, leading to significant information waste. This gap mainly arises from data characteristics: low-quality samples are often simple or repetitive with limited learning signals, while high-quality samples involve complex reasoning or rich knowledge, making them more beneficial for training (Morishita et al., 2024), as shown in Figure 1. Moreover, in many specialized domains or low-resource settings (e.g., low-resource machine translation and domain-specific tasks such as law or medicine), the scarcity of high-quality data is widely regarded as a key bottleneck that limits further progress (Abdalla et al., 2025; Dehouck & Gómez-Rodríguez, 2020; Alzubaidi et al., 2023). Recently, some works have begun exploring ways to enhance low-quality data to unlock their potential; however, most existing approaches still rely on heuristic rules or handcrafted

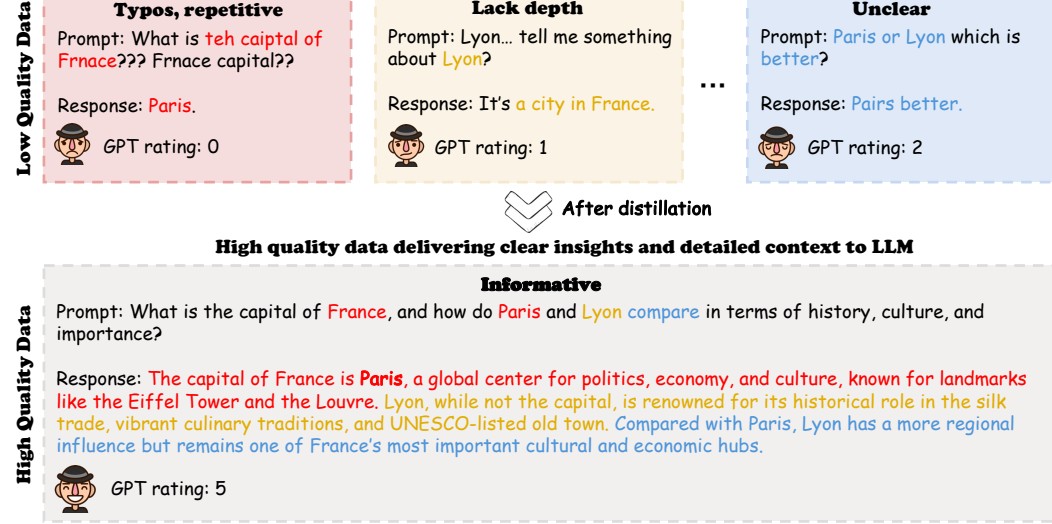

Figure 1: The goal of *Instruction Distillation*. Low-quality samples show issues such as typos, lack of depth, and unclear intent, each receiving low GPT ratings. After distillation, they are combined and refined into a single high-quality sample with clear, informative, and context-rich content. Ratings are on a 5-point scale. Additional case studies are provided in Appendix J.

templates, struggling to substantially enrich their information content or complexity (Chai et al., 2025; Zhu et al., 2025; Lee et al., 2024). This raises a key question: ***can we fully exploit low-quality data and transform it into a valuable resource for improving LLM training?***

In this work, we study how to efficiently leverage low-quality data and introduce the *Instruction Distillation* paradigm: given topic-related but sparse and incomplete inputs, the goal is to aggregate and rewrite them into a single information-dense target. To facilitate this paradigm, we construct MIXTURE, a Wikipedia-based dataset with about 144K instances across five task types, providing hierarchical mappings from multiple low-quality inputs to a single high-quality output, as shown in Figure 2. Each high quality data pair with 2 to 20 controlled low quality variants and optional chain-of-thought supervision. To further improve diversity in the dataset and robustness during the training process, cross-topic mixing and noise injection are added.

Since SFT concentrates on memorizing answers (Li et al., 2025b; Chu et al., 2025) and fails to explore diverse strategies for distilling low-quality samples into high-quality outputs, we adopt GRPO (Guo et al., 2025) to optimize the generation process. Building on MIXTURE, we further train *LM-Mixup* with GRPO to fully leverage its potential. Concretely, we perform cold-start pretraining on a subset of MIXTURE to equip the model with the basic ability to generate high-quality outputs; then, we apply GRPO-based reinforcement learning to jointly optimize output quality along three dimensions: quality, semantic alignment, and format compliance. *LM-Mixup* significantly outperforms SFT and selective baselines across multiple tasks, with small-scale models even surpassing strong instruction models under direct prompting; moreover, a small amount of original high-quality data combined with distilled results from the low-quality data (totally 10K) can achieve or exceed the performance of large-scale datasets (300K) and advanced data selection methods, demonstrating excellent data efficiency and generalization.

Our contributions are summarized as follows:
- We introduce the ***Instruction Distillation* task**, which aims to transform sparse, incomplete, and low-quality inputs into a single information-dense output; to support this paradigm, we construct MIXTURE, a **144K-instance Wikipedia-based dataset** with hierarchical mappings from multiple low-quality variants to high-quality targets.
- We introduce *LM-Mixup*, initialized through cold-start pretraining and optimized with **GRPO-based reinforcement learning** using **multi-dimensional rewards** (quality, semantic alignment, and format compliance), achieving **superior performance on the MIXTURE test set** compared to SFT and strong baselines.
- Experiments show that training downstream models on the distilled data together with the original high-quality data, totaling only ≈**3% of the full dataset**, matches or surpasses full-dataset training

Figure 2: Overview of the MIXTURE construction pipeline. The process consists of two stages: (1) Raw data collection from Wikipedia, including segmentation and prompt-based generation of five task types; (2) High- and low-quality data construction via LLM-based scoring, multi-variant degradation, cross-topic fusion, and noisy injection, producing the final dataset for training.

and advanced data selection methods on public benchmarks, demonstrating the **value of low-quality data after distilling**.

## 2 RELATED WORK

**Data-centric AI.** Recent work on data-centric AI emphasizes improving model performance through high-quality data (Ng et al., 2021; 2022). These efforts can be broadly categorized as follows: data cleaning methods (Geerts et al., 2013; Krishnan et al., 2016; 2017; Zhang et al., 2021; Sen et al., 2022; Côté et al., 2024; Skjerve et al., 2025), such as the probabilistic model Holoclean (Rekatsinas et al., 2017) and the automated framework by Mavrogiorgos et al. (2022).; Data augmentation expands training sets through strategies such as linear interpolation, geometric transformations of images (Zhang et al., 2017), or introducing latent variables (Jiang & Mei, 2019) to enrich the parameter space; Human-led data labeling and annotation, often assisted by large language models, resulting in high-quality datasets (Wang et al., 2018; Rajpurkar et al., 2016). Recent data quality scoring systems like DS2 (Pang et al., 2025) correct LLM rating errors effectively.; and data healing, which employs techniques such as model proposals, regularization adjustments and so on (Han et al., 2018; Yao et al., 2018; Tanno et al., 2019; Hu et al., 2019; Liu & Guo, 2020; Ma et al., 2020).Wang et al. (2024) developed NoiseGPT to detect and correct mislabeled instances, while Yang et al. (2024) utilized cubic regularization to efficiently identify noise. Concurrently, MergeIT (Cai et al., 2025) focuses on merging similar medium-quality instructions, whereas we study distillation from multiple noisy low-quality responses. Although low-quality data is pervasive in real-world applications, research on systematically improving its quality remains limited, and our work aims to fill this gap.

**Mixup methods.** Linear interpolation-based data augmentation has demonstrated significant advantages in enhancing model robustness and generalization across various domains (Zhang et al., 2020; Cao et al., 2024). These approaches have been adapted to textual data at multiple representation levels (Zhang & Vaidya, 2021; Chen et al., 2022). Previous work has explored interpolation in the word embedding space or sentence embedding space (Guo et al., 2019; Guo, 2020; Kong et al., 2022). Recent work has developed diverse methods to improve augmented data quality, including structural approaches like subtree decomposition in syntactic and semantic trees (Zhang et al., 2022),input-level tuning (Yoon et al., 2021), representation-level mixing of embeddings and latent features (Chen et al., 2020; Jindal et al., 2020) and so on (Yoon et al., 2021; Yang & Xiang, 2024; Zheng et al., 2023). Notably, Sun et al. (2020) were the first to directly apply mixup to textual data by fine-tuning Transformer models, enabling linear interpolation between text samples for effective augmentation.In this work, we leverage the mixup concept to systematically combine multiple low-quality, same-topic data samples into higher-quality instances, thereby enabling more effective utilization of abundant but noisy data resources.

## 3 MIXTURE: A DATASET FOR *Instruction Distillation*

### 3.1 TASK FORMULATION

The task of *Instruction Distillation* aims to aggregate and distill multiple potentially imperfect inputs (e.g., redundant or low-quality data) into a single, high-quality instruction–response pair. Let

$\mathcal{X}$ denote the universe of low-quality samples and $\mathcal{Y}$ the set of high-quality texts. Each training instance consists of a multi-source input $X = \{\ell_1, \ldots, \ell_k\} \subset \mathcal{X}$ describing the same topic/task and a reference high-quality target $Y \in \mathcal{Y}$ that is a single high-quality instruction–response pair drawn from the same task. We denote the training corpus as $\mathcal{D} = \{(X_i, Y_i)\}_{i=1}^n$. The objective is to learn a generator $f_\theta$ that semantically fuses the multi-source inputs and produces a high-quality text $\hat{Y} = f_\theta(X)$ which (i) preserves the salient information in $X$ while denoising conflicts and increasing information density, (ii) aligns semantically with $X$, and (iii) adheres to the task-specific format. Compared with standard instruction tuning, the mapping is one-to-many: there exist *multiple* valid fusions for the same $X$, reflecting an underspecified target space that requires modeling output diversity while enforcing quality.

## 3.2 DATASET STATISTICS

We introduce MIXTURE, a dataset specifically designed for *Instruction Distillation*, comprising five task types: QA pairs, True/False (TFQ), Paragraph, Multiple-Choice Question (MCQ), and Category-Statement (CS). The overall pipeline is illustrated in Figure 2. Overall, Mixture comprises 144,884 samples spanning these five task types, with a balanced distribution across normal, cross-topic, and noisy variants, as shown in Appendix K.

## 3.3 RAW DATA COLLECTION

**Source Selection.** As shown in Step 1 of Figure 2 , we use the Wikipedia dataset[1] as the initial source and sample entries across topical categories to construct the original corpus with broad coverage. To improve quality, we remove overly short entries, extract only plain text, and apply basic deduplication to eliminate redundant content. After filtering, about 10,000 Wikipedia entries are retained.

**Paragraph Segmentation.** Since Wikipedia articles are often long, directly feeding them into LLMs may cause inefficiency and instability (Liu et al., 2023a). We segment each article into semantically coherent blocks by first splitting into sentences and then greedily concatenating them until a token limit $T$[2] is reached. To balance coherence and boundary effects, we allow optional overlaps and merge very short segments, while over-length sentences are further split at punctuation marks. The final blocks preserve the original order to ensure narrative consistency and traceability.

## 3.4 HIGH- AND LOW-QUALITY DATA CONSTRUCTION

**High-Quality Sample Generation.** As shown in Step 2 of Figure 2, to transform generic Wikipedia paragraphs into task-specific samples, we use a prompt-based rewriting approach with the `ChatGPT-4o-mini` (Achiam et al., 2023). All tasks follow the principle of information density to produce knowledge-rich outputs, with MCQ, TFQ, and CS pairs differing only in prompt formats (see Appendix I). For paragraph-level tasks, segmented text blocks that are correctly parsed and meet template constraints are directly used as high-quality samples; invalid ones are discarded.

Following previous work (Li et al., 2024a; Chen et al., 2025; Pang et al., 2024), we further use `ChatGPT-4o-mini` to perform quality scoring along multiple dimensions, including rarity, complexity, and informativeness. The scores are aggregated into a single overall rating, which is then discretized to a 1-5 scale. To ensure consistency across tasks and sessions, we retain only samples with a score of 4 or above as the final high-quality subset, while those below 4 are regarded as low-quality data.

**Low-Quality Sample Generation.** To enrich alignment signals, we use `ChatGPT-4o-mini` rewriting to generate multiple degraded variants for each high-quality sample, reducing information density or reasoning completeness while preserving the topic. For each target $Y$, we construct $k \in \{2, \ldots, 20\}$ variants $X$, with the distribution shown in Appendix (Figure 9), forming hierarchical mappings that teach the model to aggregate information and complete reasoning. We further add chain-of-thought traces as intermediate supervision to improve interpretability.

---

[1] https://huggingface.co/datasets/lucadiliello/english_wikipedia
[2] We use `cl100k_base` tokenizer.

Figure 3: Overview of the training pipeline. The process involves cold-start pretraining on a subset of MIXTURE, followed by policy optimization using multi-dimensional rewards.

**Robustness Enhancement.** To enhance data diversity and robustness, we perform cross-topic synthesis by selecting semantically similar sample pairs with constrained entity overlaps, followed by GPT-based topic fusion rewriting, retaining only samples with quality scores above 4. To further improve generalization to noisy inputs, we inject surface-level perturbations such as spelling variations, synonym substitutions, and minor formatting shifts during training, while preserving some clean samples to balance robustness and fidelity.

## 4 *LM-Mixup*: TRAINING FRAMEWORK FOR *Instruction Distillation*

This section is organized into three parts: (i) Cold Start Pretraining, (ii) Multi-Dimensional Reward Design, and (iii) Reinforcement Learning with GRPO. The overall framework of the proposed *LM-Mixup* training pipeline, built upon the Qwen-2.5-1.5B-Instruct (Team, 2024), is illustrated in Figure 3.

### 4.1 COLD START

Directly starting reinforcement learning from randomly initialized parameters often leads to training instability (Guo et al., 2025; Wei et al., 2025). Therefore, we first perform cold start pretraining on the subset of MIXTURE. Specifically, given a high-quality sample $Y$ and its corresponding $k$ low-quality samples $\{\ell_1, \ell_2, \ldots, \ell_k\}$, we linearize them into a conditional input sequence

$$X = \text{Linearize}(\ell_1, \ldots, \ell_k), \tag{1}$$

and minimize the conditional likelihood with a standard autoregressive language modeling objective:

$$\mathcal{L}_{\text{CE}}(\theta) = -\sum_{t=1}^{T} \log p_\theta(y_t \mid y_{<t}, X), \tag{2}$$

where $y_t$ denotes the $t$-th token of the target output $Y$. Through this stage, the model acquires basic language generation and information fusion capabilities, providing a stable initial policy distribution for subsequent reinforcement learning.

### 4.2 REWARD DESIGN

To encourage the model to produce outputs with stronger information aggregation, semantic alignment, and structural conformity when mapping multiple low-quality samples to high-quality outputs, we design three complementary reward components. Given a model output $\hat{Y}$ and the corresponding high-quality reference $Y$, the total reward is defined as

$$R(\hat{Y}, Y) = \lambda_q R_q(\hat{Y}) + \lambda_a R_a(\hat{Y}, Y) + \lambda_f R_f(\hat{Y}), \tag{3}$$

where $\lambda_q, \lambda_a, \lambda_f$ are the normalized weights. In our experiments, we set $\lambda_q = 0.5$, $\lambda_a = 0.4$, and $\lambda_f = 0.1$.

**(1) Quality Reward $R_q$:** To efficiently approximate LLM ratings during training, we introduce a KNN-Bayes scoring scheme. Given a generated output $\hat{Y}$, we retrieve its $k$ nearest neighbors from

a large reference set with pre-computed LLM scores and estimate the posterior distribution of the true quality label via a score transition matrix $T$ (Zhu et al., 2021; Pang et al., 2024):

$$P(y = i \mid \mathbf{h}(\hat{Y})) \propto p_i \cdot \exp\left(\sum_j h_j(\hat{Y}) \log T_{ij}\right), \tag{4}$$

where $\mathbf{h}(\hat{Y})$ is the neighbor rating histogram. The expected quality score $\hat{s}(\hat{Y})$ from this posterior is then mapped into a parameterized piecewise reward:

$$R_q\big(\hat{Y}; \lambda, \kappa, \alpha, \beta\big) = \begin{cases} \alpha & \hat{s}(\hat{Y}) \geq \lambda, \\ \beta & \hat{s}(\hat{Y}) = \kappa, \\ 0 & \text{otherwise}, \end{cases} \tag{5}$$

By default we set $\lambda=4$, $\kappa=3$, and $\alpha=1$, $\beta=0.3$. The offline construction of the reference set and transition matrix estimation is provided in Appendix C.2.

**(2) Semantic Alignment Reward $R_a$:** To ensure semantic consistency between generated outputs and reference answers, we encode both using the embedding model[3] and compute the normalized cosine similarity

$$R_a(\hat{Y}, Y) = \mathbb{K}\left(\text{cosine}(e(\hat{Y}), e(Y)) \geq \tau\right), \tag{6}$$

where $e(\cdot)$ denotes the SentenceBERT encoder, $\tau$ is the similarity threshold, and $\mathbb{K}(\cdot)$ is the indicator function that returns 1 if the condition holds and 0 otherwise.

**(3) Format Compliance Reward $R_f$:** To enforce structural consistency with the `<think>...</think><answer>...</answer>` template, we use regular expressions to verify the output format. Outputs fully matching the template receive $R_f(\hat{Y}) = 1$, otherwise 0.

Finally, the total reward in Eq. equation 3 integrates quality, semantic alignment, and format compliance into a unified multi-dimensional signal.

### 4.3 REINFORCEMENT LEARNING WITH GRPO

Building on cold-start pretraining and designed rewards, we adopt GRPO for reinforcement-learning fine-tuning. Unlike standard SFT, which forces the model to imitate a single reference answer, our *Instruction Distillation* task allows infinitely many valid aggregation or generation strategies. Sole reliance on SFT risks overfitting to one canonical form and ignoring the diverse space of high-quality outputs. In contrast, reinforcement learning enables optimizing directly against reward signals, encouraging exploration of diverse outputs and progressively improving generation quality.

Specifically, GRPO is a variant of PPO (Schulman et al., 2017) that removes the need for a learned value (critic) function by replacing the baseline with group-wise statistics. For each input $X$, the model samples multiple candidate outputs $\{\hat{Y}_1, \ldots, \hat{Y}_m\}$, which are scored by the multi-dimensional reward $R(\hat{Y}_i, Y)$. To reduce variance and mitigate scale inconsistency across candidates, GRPO computes a normalized reward within each group:

$$\tilde{R}_i = \frac{R(\hat{Y}_i, Y) - \mu_X}{\sigma_X + \epsilon}, \tag{7}$$

where $\mu_X$ and $\sigma_X$ are the mean and standard deviation of $\{R(\hat{Y}_j, Y)\}_{j=1}^m$, and $\epsilon$ is a small constant to ensure numerical stability. The policy optimization objective becomes:

$$\mathcal{L}_{\text{GRPO}}(\theta) = \mathbb{E}_X\left[\min\big(r_i(\theta)\,\tilde{R}_i,\, \text{clip}(r_i(\theta), 1 - \epsilon_{\text{clip}}, 1 + \epsilon_{\text{clip}})\,\tilde{R}_i\big)\right] - \beta\,\text{KL}\big(\pi_\theta(\cdot \mid X) \,\|\, \pi_{\theta_0}(\cdot \mid X)\big), \tag{8}$$

where $r_i(\theta) = \frac{\pi_\theta(\hat{Y}_i|X)}{\pi_{\theta_0}(\hat{Y}_i|X)}$ is the importance ratio between the current policy $\pi_\theta$ and the reference (old) policy $\pi_{\theta_0}$; $\epsilon_{\text{clip}}$ is the PPO clipping parameter; and $\beta$ controls the strength of the KL regularization to ensure stability (Christiano et al., 2017).

---

[3] https://huggingface.co/BAAI/bge-m3

Table 1: Performance comparison across different models on five tasks. The best results per column are highlighted in **bold**.

| Model | cs | mcq | para | qa | tfq | Avg |
|---|---|---|---|---|---|---|
| LLaMA-3.1-8B-Instruct | 3.61 | 2.57 | 3.57 | 3.71 | 2.10 | 3.27 |
| LLaMA-3.2-3B-Instruct | 3.58 | 2.66 | 3.52 | 3.78 | 2.49 | 3.21 |
| DeepSeek-R1-Distill-Qwen-7B | 3.61 | 2.46 | 3.40 | 3.23 | 2.41 | 3.02 |
| Qwen-2.5-7B-Instruct | 3.70 | 2.77 | 3.58 | 3.53 | 2.57 | 3.28 |
| Qwen-2.5-1.5B-Instruct | 3.39 | 2.44 | 3.34 | 3.33 | 1.34 | 2.86 |
| GPT-4o-mini | 3.81 | 2.86 | **3.69** | 3.64 | 2.61 | 3.37 |
| Qwen-2.5-1.5B-SFT | 3.54 | 3.25 | 2.82 | 3.73 | 3.05 | 3.28 |
| Qwen-2.5-7B-SFT | 3.53 | 3.31 | 3.41 | 3.78 | 3.10 | 3.46 |
| *LM-Mixup* | **3.85** | **3.55** | 3.31 | **4.17** | **3.32** | **3.66** |

Table 2: Data pool statistics.

| Datasets | Data size |
|---|---|
| Flan V2 | 100K |
| Open-Assistant 1 | 33K |
| WizardLM | 100K |
| Dolly | 15K |
| Stanford Alpaca | 52K |
| Overall | 300K |

## 4.4 CAPACITY-CONSTRAINED CLUSTERING

After GRPO training, the model can distill multiple low-quality samples into high-quality ones. For downstream tasks, we introduce a Capacity-Constrained Clustering method to automatically collect low-quality inputs with flexible control over cluster number and size, which also mitigates the severe imbalance or over-fragmentation issues often observed in standard clustering methods. Given a text collection $\mathcal{D} = \{x_i\}_{i=1}^{N}$, we encode each sample into $\mathbf{h}_i \in \mathbb{R}^d$ using a pre-trained encoder. A target capacity vector $\mathbf{c} = (c_1, \ldots, c_K)$ is drawn from a truncated normal distribution with $c_k \in [c_{\min}, c_{\max}]$. We then perform two-stage clustering: (i) run MiniBatchKMeans to obtain $k$ initial cluster centers $\{\mathbf{c}_k\}$; (ii) iteratively assign samples to the most similar clusters under capacity constraints, with a few refinement steps to ensure semantic compactness and balanced partitioning.

## 5 EXPERIMENTS

### 5.1 MIXTURE EXPERIMENTAL RESULTS

**Experimental Setup.** To comprehensively evaluate the performance of *LM-Mixup* on the MIXTURE dataset, we conducted standardized experiments on the test set using a variety of models. Specifically, the experiments involved the following models: ChatGPT-4o-mini (Achiam et al., 2023), Qwen-2.5-1.5B-Instruct, Qwen-2.5-7B-Instruct (Team, 2024), LLaMA-3.1-8B-Instruct, LLaMA-3.2-3B-Instruct (Dubey et al., 2024), DeepSeek-R1-Distill-Qwen-7B (Guo et al., 2025), Qwen-2.5-1.5B-SFT, Qwen-2.5-7B-SFT (obtained via supervised fine-tuning on the full MIXTURE). All models were evaluated on the same test set, constructed by holding out a non-overlapping 20% split from MIXTURE, under identical prompt conditions to ensure fair comparison. For automated evaluation, we employed ChatGPT-4o-mini as the rating model to assess the quality of the generated outputs.

**Results.** Table 1 presents the performance comparison across different models on the MIXTURE test set. It can be observed that *LM-Mixup* consistently outperforms all baseline models, achieving the best overall results. Compared with standard supervised fine-tuning, GRPO training with multi-dimensional quality rewards enables the model to learn generation patterns that produce higher-quality answers rather than merely mimicking the ground truth.

### 5.2 OPENLLM LEADERBOARD EVALUATION RESULTS

**Experimental Setup.** Following previous work (Pang et al., 2024), to evaluate *LM-Mixup* 's performance on OOD datasets, we construct an additional data pool consisting of Flan_v2 (Longpre et al., 2023), Open Assistant 1 (Köpf et al., 2023), WizardLM (Xu et al., 2023), Dolly (Conover et al., 2023), and Stanford Alpaca (Taori et al., 2023). Detailed statistics of the data pool are provided in Table 2. To identify low-quality samples within this pool, we employ `ChatGPT-4o-mini` for quality rating following the same protocol described in Sec 3.4, where samples with a score below 4 are regarded as low-quality data. We then apply *LM-Mixup* to perform mixup on the low-quality samples within the data pool and compute long-tail scores using embeddings. The top-ranked samples from both the original high-quality data and the mixup data from low-quality data are then selected for instruction fine-tuning.

Table 3: Results on the OpenLLM leaderboard using `LLaMA-3.1-8B` as the base model. The top-performing scores are shown in **bold**, while the second-best scores are marked with underlines. Unless otherwise specified, the size of the fine-tuning dataset is 10K. * indicates that the values are sourced from Pang et al. (2024).

| Model | MMLU (factuality) | TruthfulQA (truthfulness) | GSM (reasoning) | BBH (reasoning) | TydiQA (multilinguality) | Average |
|---|---|---|---|---|---|---|
| VANILLA BASE MODEL* | 64.1 | 33.5 | 56.5 | 55.4 | 23.3 | 46.6 |
| COMPLETION LENGTH* | 64.2 | 41.4 | 62.5 | 60.7 | 23.0 | 50.4 |
| PERPLEXITY* | 63.1 | 40.4 | 55.5 | 60.2 | 62.1 | 56.3 |
| $k$-NN-10* | 62.4 | 44.3 | 57.0 | 59.1 | 63.8 | 57.3 |
| RANDOM SELECTION* | 63.4 | 39.1 | 62.2 | 61.3 | 61.1 | 57.4 |
| LESS* | 63.0 | 39.0 | 57.5 | 63.1 | 67.2 | 58.0 |
| FULL DATA (300K)* | 63.5 | 42.0 | 61.0 | 59.1 | 62.8 | 57.7 |
| ALPAGASUS* | 63.4 | 42.6 | 66.0 | 59.1 | 59.4 | 58.1 |
| DEITA* | 64.5 | 50.1 | 60.0 | 60.3 | 63.7 | 59.7 |
| DS2 W/O CURATION* | 63.3 | 51.5 | 62.0 | 59.7 | 64.3 | 60.2 |
| DS2* | 64.0 | 50.3 | 67.5 | 59.0 | 66.1 | 61.4 |
| BACK-TRANSLATION | 62.0±0.4 | 46.5±2.9 | 61.2±0.8 | 58.8±2.2 | 60.2±0.8 | 57.7±0.1 |
| EDA | 61.6±0.9 | 43.7±2.0 | 56.2±1.0 | 59.7±0.3 | 62.0±1.6 | 56.6±0.6 |
| REPHRASING | 61.4±0.7 | 36.0±2.5 | 63.2±1.0 | 59.6±0.2 | 62.2±0.8 | 56.5±0.6 |
| BASE 70% + ORI 30% | 62.2±0.9 | 40.7±0.5 | 54.3±0.2 | 55.1±1.1 | 23.2±0.3 | 47.1±0.1 |
| BASE 50% + ORI 50% | 62.1±0.1 | 37.4±0.7 | 50.8±0.6 | 54.1±0.7 | 22.9±0.4 | 45.4±0.2 |
| BASE 30% + ORI 70% | 61.3±0.7 | 38.2±0.7 | 51.2±1.4 | 54.2±0.9 | 23.0±0.5 | 45.6±0.4 |
| LOW 70% + ORI 30% | 62.7±0.7 | 17.8±2.0 | 62.5±4.0 | 60.3±0.9 | 65.6±1.1 | 53.6±1.2 |
| MIXUP 70% + ORI 30% | 63.0±0.2 | 47.9±0.3 | 63.3±0.6 | 61.1±0.3 | 64.2±0.6 | 59.9±0.1 ↑6.3 |
| LOW 50% + ORI 50% | 62.4±0.6 | 39.0±9.8 | 62.7±1.1 | 61.0±2.2 | 64.0±0.3 | 57.9±2.0 |
| MIXUP 50% + ORI 50% | 63.3±0.3 | 52.6±0.1 | 65.5±0.2 | 61.3±0.3 | 64.6±0.3 | 61.5±0.1 ↑3.6 |
| LOW 30% + ORI 70% | 60.9±2.1 | 41.1±5.6 | 62.7±1.9 | 59.9±1.7 | 60.4±1.7 | 57.0±0.8 |
| MIXUP 30% + ORI 70% | 63.1±0.3 | 46.8±0.6 | 61.2±1.5 | 58.0±0.2 | 63.4±1.1 | 58.5±0.1 ↑1.5 |

**Metrics.** We report task-specific metrics, including accuracy on MMLU (Hendrycks et al., 2020), BBH (Suzgun et al., 2022), and GSM8K (Cobbe et al., 2021), the Informative-Truthful Rate on TruthfulQA (Lin et al., 2021), and F1 scores on TyDiQA (Clark et al., 2020).

**Training Settings.** We fine-tune three base models, LLaMA-2-7B (Touvron et al., 2023), LLaMA-3.1-8B (Dubey et al., 2024), and Mistral-7B-v0.3 (Jiang et al., 2023), on 10K samples under six settings: three with *mixup* data, three with *direct* low-quality samples (without mixup), each combined with original high-quality data at 70%, 50%, or 30%. For all experiments newly conducted by us, we report the average results over three independent runs to ensure statistical reliability, whereas the baseline results sourced from prior work are reported as originally published. Notably, for the full original data setting, we adopt DS2 (Pang et al., 2024) as the baseline for comparison. Additionally, we also provide the results for both the full mixup data and the full low-quality data in Appendix G. Further details of the additional training settings are provided in Appendix F.

**Baselines.** We compare our method against several representative data selection baselines commonly used in LLM fine-tuning, including *Random Selection*, *Completion Length*, *Perplexity*, *k-NN*, *LESS* (Xia et al., 2024), *AlpaGasus* (Chen et al., 2023b), *DEITA* (Liu et al., 2023b), *DS2* (Pang et al., 2024), and *Full Data*. We also include widely used data augmentation baselines, including backtranslation (Edunov et al., 2018), paraphrasing (Abaskohi et al., 2023), and EDA (Wei & Zou, 2019). Additionally, to more comprehensively assess the gains achieved by *LM-Mixup*, we also report the zero-shot performance of Qwen-1.5B-Instruct without any further training. Detailed descriptions of these baselines are provided in Appendix E.

**Low-quality data matters: After being processed by *LM-Mixup*, it can even outperform high-quality-only baselines.** Table 3 shows that combining low-quality data (after mixup) with original high-quality samples can surpass baselines that rely solely on high-quality data selection. In particular, the 50% mixup + 50% original configuration achieves the top average score across all five OpenLLM Leaderboard benchmarks, with Tables 5 and 6 showing similar trends on Mistral-7B and LLaMA-2-7B. This demonstrates that even low-quality data, when fused into high-quality samples, can enhance diversity and complement real data to boost performance. Additional results on more models are provided in the Appendix G.

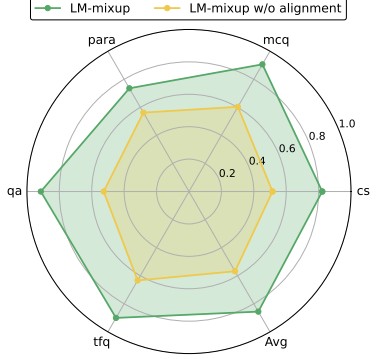 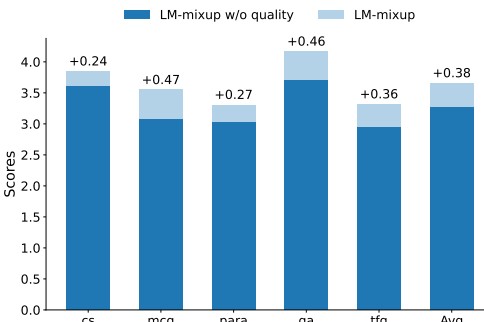

Figure 4: Ablation study on reward components in *LM-Mixup*. The left figure evaluates the effect of removing the alignment reward, while the right figure shows the impact of removing the quality reward across different tasks.

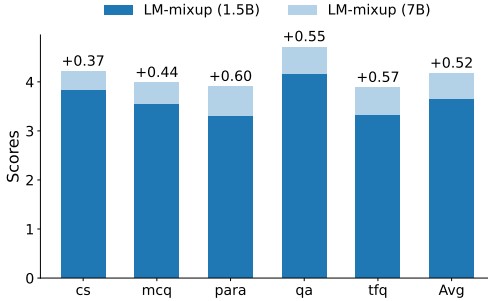

Figure 5: Effect of model scaling on performance.

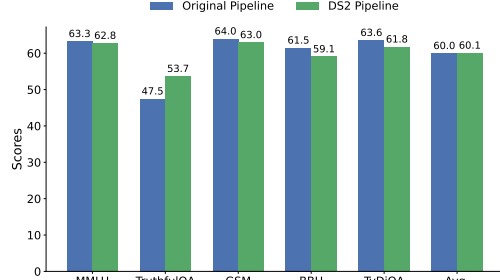

Figure 6: Comparison of the *LM-Mixup* using DS2 pipelines under the Mixup 70% + Ori 30%.

**LM-Mixup demonstrates strong performance.** As shown in Table 3, *LM-Mixup* significantly outperforms the zero-shot Qwen-1.5B-Instruct model used for mixup without any training. Furthermore, our method consistently surpasses standard data augmentation baselines, indicating that it truly elevates low-quality samples into high-quality supervision signals rather than merely increasing surface-level diversity.

**Mixup outperforms the full data pool with only 3.3% of the data.** Notably, on both LLaMA-3.1-8B (Table 3) and Mistral-7B (Table 5), our best-performing mixup configuration using only 10K training samples even surpasses the 300K full data pool baseline, demonstrating that mixup not only enhances data diversity but also enables a highly compact training set to outperform large-scale unfiltered data.

**Effect of mixup on low quality data.** As shown in Table 3, applying *LM-Mixup* to low-quality data consistently improves performance across all mixture ratios. E.g., in the 70% low-quality + 30% high-quality setting, *LM-Mixup* raises the score from 54.2 to 60.0 (↑5.8), with similar gains in the 50% (↑1.6) and 30% (↑2.2) settings. This highlights that properly modeling low-quality data can yield substantial benefits for model training.

## 5.3 ABALTION STUDY

**Ablation on Reward Components.** We conduct ablation studies to investigate the contribution of each reward component in our GRPO-based *LM-Mixup* training framework, which incorporates quality and alignment rewards alongside the base objective. As shown in Figure 4, removing the alignment reward causes the model to exhibit reward hacking behavior: it tends to memorize answers from the reference set regardless of the input, leading to significantly lower semantic similarity with the ground truth. On the other hand, removing the quality reward makes the model behave similarly to standard SFT, producing outputs with limited quality improvement. These results highlight that both rewards are essential: the alignment reward ensures semantic faithfulness to the input, while the quality reward drives the generation of high-quality outputs beyond simple imitation.

**Effect of Model Scaling.** To investigate the impact of scaling up model parameters, we extend our training pipeline from Qwen-2.5-1.5B-Instruct to Qwen-2.5-7B-Instruct using the same GRPO-based optimization described in Sec.4. As shown in Fig.5, the larger 7B model consistently outperforms its 1.5B counterpart across all tasks on the MIXTURE test set, achieving an average score of 4.18 compared to 3.66 on the smaller model. These results demonstrate both the effectiveness and the scalability of our approach when applied to models with larger parameter sizes.

**Revisiting LLM rating bias.** Recent work has noted that LLM-as-judge scores can be biased (Ye et al., 2024; Chen et al., 2024). In our pipeline we use `ChatGPT-4o-mini` for rating, which may introduce such bias. To assess sensitivity, we conducted experiments using the DS2 pipeline (Pang et al., 2024). We conduct the same experiments described in Sec.5.2 experiments under the Mixup 70% + Ori 30% setting, where the overall performance shows only marginal changes, as shown in Fig.6. We hypothesize two reasons: (i) *LM-Mixup* 's GRPO with multi-dimensional rewards and many-to-one mixup supervision provides strong signals that attenuate upstream rating noise; and (ii) diversity is governed by embedding-based long-tail selection, largely independent of the rating scale. Overall, while LLM rating bias is real, our design appears tolerant to moderate bias; further de-biasing (e.g., multi-judge ensembling, cross-model adjudication, or light human spot-checks) may be needed to unlock additional gains.

# 6 CONCLUSION

In this work, we introduce *Instruction Distillation* and present a comprehensive data construction pipeline to create MIXTURE, a large-scale dataset pairing low-quality, noisy and redundant instruction clusters with their high-quality distillations. Building on this MIXTURE, we propose *LM-Mixup*, a model trained with supervised fine-tuning followed by reinforcement learning using customized rewards. Our results demonstrate that: *LM-Mixup* can efficiently distill plenty of imperfect data samples into condensed high-quality ones, significantly compress the training data size, fully extract the information value of neglected low quality data, and meanwhile effectively enhance the efficiency and performance of instruction-tuned LLMs.

ETHICS STATEMENT

We use both publicly available datasets and our constructed MIXTURE derived from Wikipedia text under its open license, ensuring no sensitive or private information is included. LM-Mixup distills low-quality or redundant samples into high-quality data while filtering harmful content, reducing data scale and computational cost to support responsible and sustainable AI development.

REPRODUCIBILITY STATEMENT

Our experimental settings are detailed in Section 5 and Appendix D, and both the code and dataset will be released upon paper acceptance.

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

# APPENDIX

## A   THE USE OF LARGE LANGUAGE MODELS

In this work, we employ ChatGPT-5, a state-of-the-art large language model, to assist with language refinement and clarity improvement. Specifically, ChatGPT-5 is used to polish the writing style, correct grammatical errors, and enhance the overall readability of the manuscript without altering its scientific content or conclusions.

## B   CONCEPTUAL DISTINCTION FROM DATA AUGMENTATION AND DATA CURATION

A central goal of our work is to address the pervasive scarcity of high-quality supervision in many NLP scenarios, especially for low-resource or domain-specific tasks. In such settings, large quantities of noisy, redundant, or otherwise low-quality instruction–response pairs are often available, whereas carefully curated high-quality data are expensive and limited. Our instruction distillation framework aims to transform abundant low-quality signals into a compact set of high-quality supervision examples, thereby improving the utility of existing corpora for downstream instruction tuning and supporting the broader low-resource NLP community.

Concretely, the MIXTURE dataset defines five heterogeneous task types. As shown in Sec. 5.1, LM-mixup achieves consistently strong performance across these tasks, indicating that the proposed instruction distillation paradigm is not tied to a single setting or a narrow engineering trick, but rather exhibits robustness and generality across diverse instruction-following scenarios.

Conceptually, our notion of *Instruction Distillation* is distinct from both traditional data augmentation and data curation:

- **Instruction distillation.** Given multiple low-quality or inconsistent responses, instruction distillation extracts the useful information across them and semantically fuses these weak signals into a single, high-quality instruction–response pair. This process simultaneously aggregates information, denoises spurious content, and enforces quality-control constraints, yielding supervision with higher information density for downstream models.

- **Data augmentation.** Classical augmentation techniques generate additional samples via transformations such as rewriting (Wei & Zou, 2019), paraphrasing (Abaskohi et al., 2023), back-translation (Edunov et al., 2018), or other synthetic procedures, with the primary goal of expanding data volume and increasing diversity and robustness (Feng et al., 2021; Chen et al., 2023a). The underlying semantic content of each example is usually preserved, and the number of samples grows.

- **Data curation.** Data curation typically focuses on improving annotation quality or consistency for existing samples without substantially changing their semantic content (Northcutt et al., 2021; Karimi et al., 2020). Examples include relabeling noisy instances, filtering problematic examples, or correcting minor errors while keeping the original instruction–response structure intact.

Instruction distillation fundamentally differs from these two paradigms in both direction and effect. Instead of increasing the number of samples, it *reduces* data volume via information aggregation, while *changing and enriching* the semantic content through fusion across multiple weak sources. Unlike augmentation, which primarily improves diversity, or curation, which mainly refines labels for fixed content, instruction distillation explicitly converts many low-quality signals into a few high-information-density instructions. This enables substantial gains in low-resource regimes, where the key bottleneck is not the absolute number of examples, but the lack of sufficiently rich and reliable supervision.

# C   DETAILS OF KNN–BAYES RATING

## C.1   KNN–BAYES QUALITY MODELING WITH SCORE TRANSITION MATRIX

In the Sec.4.2, we introduced an offline KNN–Bayes calibration method to approximate the original LLM ratings during training. Intuitively, given the $k$-nearest neighbors of each sample in the embedding space, we aim to infer its "true" quality score based on the observed ratings of these neighbors. However, the LLM-provided scores $\tilde{y}$ typically suffer from systematic noise and random fluctuations. Directly averaging the neighbor scores may therefore introduce significant bias into the reward signal.

To address this issue, we adopt the classical idea of Score Transition Matrix (STM) from weak supervision and noisy-label learning, which models the conditional distribution between observed and latent labels. Let the latent true label be $y \in \mathcal{Y} = \{1, 2, \ldots, C\}$ and the observed noisy rating be $\tilde{y}$. In our implementation, we set $C = 6$ with label set $\{0, 1, 2, 3, 4, 5\}$, which matches the original data annotation. The STM is defined as

$$T \in \mathbb{R}^{C \times C}, \qquad T_{ij} = \mathbb{P}(\tilde{y} = j \mid y = i), \tag{9}$$

where $T_{ij}$ denotes the probability that a true label $i$ is perturbed into the noisy label $j$. The prior distribution is given by

$$p \in \Delta^C, \qquad p_i = \mathbb{P}(y = i), \quad \sum_i p_i = 1. \tag{10}$$

When $T = I$, the observed ratings are noise-free; deviations of $T$ from the identity matrix characterize systematic label noise.

**$k$-NN Clusterability Assumption (Wei et al., 2020).** In the embedding space, if $x'$ belongs to the $k$-nearest neighbors $\mathcal{N}_k(x)$ of $x$, then it is more likely that $y(x') = y(x)$. Based on this assumption, the neighborhood agreement frequencies yield a set of linear equations over $(T, p)$. We adopt 2-NN consensus statistics when estimating $(T, p)$ to ensure identifiability. For the posterior computation of a single sample, we use the $k$-nearest neighbor histogram $h(x)$ with $k \geq 2$ to enhance robustness. Specifically, using pairwise or triplet neighbor agreement, we define

$$v^{[1]} = T^\top p, \qquad v_\ell^{[2]} = (T \circ T_\ell)^\top p, \qquad v_{\ell,s}^{[3]} = (T \circ T_\ell \circ T_s)^\top p, \tag{11}$$

where $T_\ell = TA_\ell$ is the cyclic shift of $T$ by $\ell$ units, and $\circ$ denotes the Hadamard product. The observed frequencies $\widehat{v}^{[1]}, \widehat{v}_\ell^{[2]}, \widehat{v}_{\ell,s}^{[3]}$ can be directly computed from data, forming a linear program over $(T, p)$. We solve for $(T, p)$ subject to $T\mathbf{1} = \mathbf{1}$, $T \geq 0$, $p \geq 0$, and $\mathbf{1}^\top p = 1$. Existing theory shows that under mild identifiability conditions, third-order consensus vectors suffice to uniquely recover $(T, p)$.

Once $(T, p)$ are estimated, given the empirical neighbor histogram $h_j(x)$ of sample $x$, the posterior distribution is computed as

$$\mathbb{P}(y = i \mid h(x)) \propto p_i \prod_{j \in \mathcal{Y}} T_{ij}^{h_j(x)} = p_i \exp\left(\sum_{j \in \mathcal{Y}} h_j(x) \log T_{ij}\right), \tag{12}$$

Here $h_j(x) \in \{0, 1, \ldots, k\}$ counts the number of neighbors whose observed label equals $j$, hence $\sum_{j \in \mathcal{Y}} h_j(x) = k$. If distance weights $w_r$ are used, we replace $h_j(x)$ by the weighted sum $\sum_{r : \tilde{y}_r = j} w_r$. This posterior relies on the conditional independence assumption: given the true label $y$, the observed ratings of neighbors are mutually independent. When $T$ is diagonally dominant (close to $I$), the posterior behavior approaches that of frequency- or average-based voting. If $T = I$ without any smoothing, however, the likelihood degenerates; thus, we apply mild smoothing to $T$ and compute in the log domain to ensure numerical stability. The posterior expectation score is

$$\hat{s}(x) = \sum_{i=1}^C i \cdot \mathbb{P}(y = i \mid h(x)). \tag{13}$$

Finally, the quality reward used in training is given by the piecewise mapping

$$R_q(x) = \begin{cases} 1, & \hat{s}(x) \geq 4, \\ 0.3, & 3 \leq \hat{s}(x) < 4, \\ 0, & \text{otherwise.} \end{cases} \tag{14}$$

As $T$ becomes diagonally dominant (i.e., $T \approx I$), the posterior concentrates on the most frequent neighbor labels and behaves like smoothed majority/frequency voting.[4] When $T$ departs from $I$, the Bayesian calibration systematically corrects label noise.

## C.2 DETAILS OF KNN–BAYES QUALITY REWARD CONSTRUCTION

To efficiently approximate LLM ratings during training, we introduce a KNN–Bayes scoring system, which leverages neighborhood information and a score transition matrix to denoise label noise. The construction consists of the following steps:

**Offline Asset Construction.** We collect approximately 100K samples rated by `ChatGPT-4o-mini` as a reference set. We build a KNN index in the embedding space and compute neighbor rating co-occurrence frequencies to estimate both the score transition matrix $T$ and the label prior $\mathbf{p}$ offline, prior to model training.

**Online Inference and Reward Computation.** During training, for each generated output $\hat{Y}$, we retrieve its $k$ nearest neighbors in the reference set to form a rating histogram $\mathbf{h}(\hat{Y}) \in \mathbb{R}^C$. We then compute the posterior distribution over true labels as

$$P(y = i \mid \mathbf{h}(\hat{Y})) \propto p_i \cdot \exp\left( \sum_j h_j(\hat{Y}) \log T_{ij} \right), \tag{15}$$

and obtain the expected score

$$\hat{s}(\hat{Y}) = \sum_{i=1}^{C} i \cdot P(y = i \mid \mathbf{h}(\hat{Y})). \tag{16}$$

Finally, rewards are assigned using a piecewise mapping:

$$R_q(\hat{Y}) = \begin{cases} 1 & \hat{s}(\hat{Y}) \geq 4, \\ 0.3 & \hat{s}(\hat{Y}) = 3, \\ 0 & \text{otherwise.} \end{cases} \tag{17}$$

## C.3 CONSISTENCY BETWEEN KNN–BAYES RATING AND LLM SCORES

**Setup.** To evaluate the effectiveness of KNN–Bayes in approximating the original LLM scores, we conduct an offline stratified experiment with a reference set and an evaluation set. Given a dataset $\mathcal{D}$ with LLM-provided scores $\tilde{y}$, we first split it into a reference set $\mathcal{B}$ and an evaluation set $\mathcal{A}$ via stratified sampling to preserve the label distribution of $\tilde{y}$ across both sets. On $\mathcal{B}$, we construct a semantic embedding index and estimate the score transition matrix $T$ and prior distribution $p$ through neighborhood co-occurrence statistics. For each sample $x \in \mathcal{A}$, we retrieve its $k$-nearest neighbors in $\mathcal{B}$, obtain the empirical histogram $h(x)$, and compute the posterior distribution via

$$\mathbb{P}(y = i \mid h(x)) \propto p_i \prod_{j \in \mathcal{Y}} T_{ij}^{h_j(x)}. \tag{18}$$

We then calculate the expected score

$$\hat{s}(x) = \sum_{i \in \mathcal{Y}} i \cdot \mathbb{P}(y = i \mid h(x)). \tag{19}$$

---

[4]We apply mild Laplace smoothing $T \leftarrow (1 - \alpha)T + \alpha \mathbf{1}\mathbf{1}^\top / C$ with small $\alpha > 0$, followed by row-wise renormalization; computations are carried out in the log domain to avoid underflow.

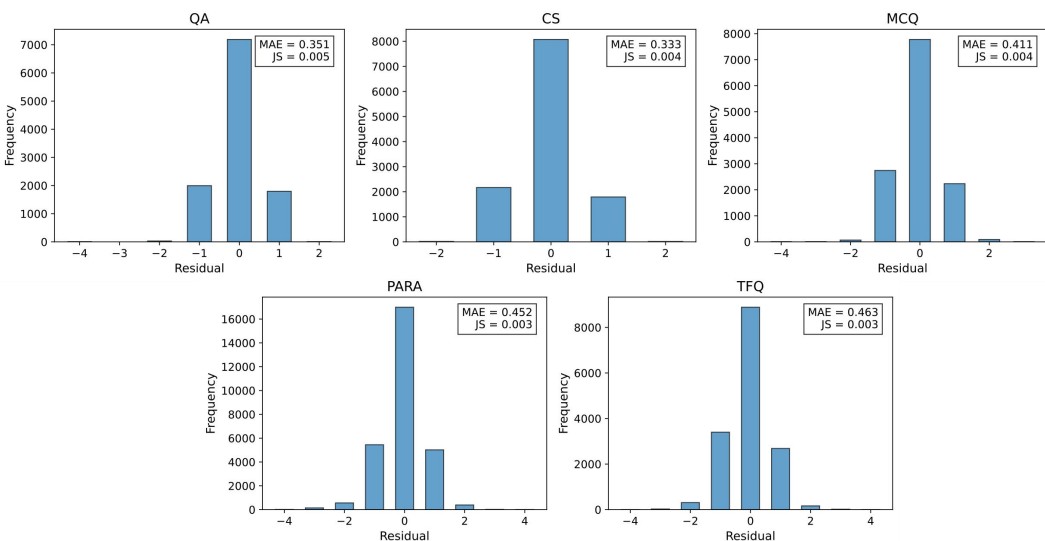

Figure 7: Residual distributions between KNN–Bayes and LLM scores on five datasets. Most residuals concentrate at zero with moderate deviations within $\{-1, 1\}$, as shown by MAE and JS metrics.

**Metrics.** We assess the consistency between KNN–Bayes scores and original LLM scores from two perspectives: distributional divergence and numerical deviation. The distributional divergence is measured by the Jensen–Shannon (JS) divergence:

$$\mathrm{JS}(P, Q) = \tfrac{1}{2}D_{KL}(P \parallel M) + \tfrac{1}{2}D_{KL}(Q \parallel M), \quad M = \tfrac{1}{2}(P + Q), \tag{20}$$

where $P$ and $Q$ denote the empirical distributions of $\hat{y}(x)$ and $\tilde{y}(x)$, respectively, with $\hat{y}(x) = \mathrm{round}(\hat{s}(x)) \in \mathcal{Y}$ (alternatively, we bin $\hat{s}(x)$ into the same $C$ categories). The numerical deviation is quantified using the Mean Absolute Error (MAE):

$$\mathrm{MAE} = \frac{1}{|\mathcal{A}|} \sum_{x \in \mathcal{A}} |\hat{s}(x) - \tilde{y}(x)|. \tag{21}$$

**Results.** Figure 7 shows the residual distributions and MAE/JS metrics across all five datasets. Overall, most residuals concentrate at 0, while some fraction falls within $\{-1, 1\}$, suggesting that KNN–Bayes captures the main structure of the original LLM ratings but still exhibits small local deviations. Quantitatively, the JS divergence remains below 0.006 on all datasets, indicating that the calibrated scores preserve the global distributional shape of the LLM scores with minimal shift. The MAE lies in the range 0.33–0.46, which is moderate compared to the discrete rating scale $\mathcal{Y} = \{0, 1, 2, 3, 4, 5\}$, reflecting that individual predictions can occasionally deviate by one score level. These findings suggest that while KNN–Bayes provides a low-cost and reasonably accurate approximation for offline evaluation.

# D  TRAINING AND EVALUATION DETAILS

## D.1  TRAINING DETAILS

We adopt a three-stage training pipeline: (1) *Cold-start full-parameter tuning*, (2) *GRPO reinforcement learning*, and (3) *Evaluation-stage fine-tuning*. All experiments are conducted on 3 H20 GPUs. The key hyperparameters for each stage are summarized below.

**Cold-start Training.** We first perform full-parameter supervised fine-tuning on the initial dataset to provide a strong initialization for later stages. This stage uses a batch size of 128, learning rate $2 \times 10^{-5}$, and runs for 3 epochs with a maximum sequence length of 2048 tokens.

**GRPO Reinforcement Learning.** The second stage adopts GRPO with multi-dimensional reward signals, including Bayesian KNN-based quality scores, BGE-M3 semantic alignment, and format regularization. We set the rollout batch size to 128, actor global batch size to 16, learning rate $1 \times 10^{-6}$, KL penalty coefficient $1 \times 10^{-2}$, and run for 1 epoch with dynamic batching and gradient checkpointing enabled.

**Evaluation-stage Fine-tuning.** Finally, following prior work (Pang et al., 2024), we perform lightweight LoRA fine-tuning with a rank size of 64 and a scaling factor of 16. We adopt a batch size of 128, a learning rate of $1 \times 10^{-4}$, and train for 5 epochs to ensure consistent settings across all evaluation benchmarks.

Table 4: Key hyperparameter settings across three training stages.

| Parameter | Cold-start | GRPO | Eval-tuning |
|---|---|---|---|
| Batch size | 128 | 128 (rollout) / 16 (actor) | 128 |
| Learning rate | $2 \times 10^{-5}$ | $1 \times 10^{-6}$ | $1 \times 10^{-4}$ |
| Epochs | 3 | 1 | 5 |
| Max sequence length | 2048 | 2048 | 2048 |
| KL penalty | – | $1 \times 10^{-2}$ | – |
| LoRA rank / scaling | – | – | 64 / 16 |
| Dropout rate | 0.1 | 0.1 | 0.1 |

## D.2 EVALUATION DETAILS

Following previous work (Pang et al., 2024), we evaluate the fine-tuned models on five widely used benchmarks: MMLU (Hendrycks et al., 2020), BBH (Suzgun et al., 2022), GSM8K (Cobbe et al., 2021), TruthfulQA (Lin et al., 2021), and TyDiQA (Clark et al., 2020). For each dataset, we follow standard protocols or common configurations. Specifically, 0-shot settings are used for MMLU; 8-shot in-context examples for GSM8K; 3-shot settings without chain-of-thought for BBH; 6-shot prompts for TruthfulQA; and one in-context example per language for TyDiQA.

## E  BASELINE DETAILS

We provide detailed descriptions of all baselines considered in the main experiments:

- **Random Selection**: Randomly selects training samples without any filtering.

- **Completion Length**: Uses the total conversation length as a proxy for data quality, assuming longer completions indicate richer information.

- **Perplexity**: Computes perplexity in a zero-shot manner using a pre-trained model; higher perplexity suggests rarer or more complex samples.

- **k-NN**: Measures average distance to the $k$ nearest neighbors in the SentenceBERT embedding space to quantify data rarity.

- **LESS** (Xia et al., 2024): Scores samples by their influence on a validation set, estimated via gradient-based metrics.

- **AlpaGasus** (Chen et al., 2023b): Employs an LLM to assign quality ratings, selecting only high-scoring samples.

- **DEITA** (Liu et al., 2023b): Scores samples by both quality and complexity, while iteratively enforcing diversity constraints.

- **DS2** (Pang et al., 2024): Selects high-quality and diverse samples by correcting LLM-generated scores via a transition matrix and combining them with long-tail diversity scores.

- **EDA** (Wei & Zou, 2019): Applies simple text-level perturbations such as synonym replacement, random insertion, random deletion, and word swapping to increase surface-level diversity without altering the original semantic content.

- **Rephrasing** (Abaskohi et al., 2023): Generates semantically equivalent paraphrases of the original instructions using GPT-4o mini, aiming to modify expression style while preserving meaning to introduce natural linguistic variation.

- **Back-translation** (Edunov et al., 2018): Translates each sample into an intermediate language and back to the source language, producing paraphrastic variants that expose the model to diverse lexical and syntactic forms.

- **Full Data**: Uses the entire dataset without any filtering for model fine-tuning.

For all rating-based methods (*AlpaGasus*, *DEITA*, and *DS2*), we follow *LM-Mixup* and adopt `ChatGPT-4o-mini` as the rating model for a fair comparison.

For all data augmentation baselines, the augmentation operations are typically applied to the entire dataset, whereas our other baselines are constructed using 10K training samples. To ensure a fair comparison, we first randomly sample 5K instances from the original 300K data pool and then apply the corresponding augmentation method to these 5K samples, resulting in a total of 10K training examples.

## F   ADDITIONAL DETAILS OF DATA PREPARATION AND EXPERIMENTAL SETTINGS

In our experiments, all 300K samples are annotated with a quality score ranging from 1 to 5. Among them, approximately 30K samples with scores $\geq 4$ are treated as high-quality data, while roughly 270K samples with scores $< 4$ constitute the low-quality pool. Below we provide additional clarifications for the datasets used in Table 3.

**ORI (High-Quality Data).**   From the 30K high-quality samples, we compute long-tail diversity scores and select the top $N$ instances to form the ORI training set. These samples serve as the high-quality-only baseline.

**LOW (Low-Quality Data).**   To construct the LOW baseline, we randomly sample $N$ instances from the 270K low-quality pool. This setting evaluates the performance of directly using weak signals without any enhancement.

**BASE (Qwen-1.5B-Instruct Generated Data).**   We directly apply the off-the-shelf Qwen-1.5B-Instruct model (without any training or fine-tuning) to fuse inputs from the low-quality pool. The generated outputs constitute the BASE dataset.

**MIXUP (LM-Mixup Generated Data).**   The MIXUP dataset is constructed using the following pipeline:

1. Sample multiple groups of low-quality inputs from the 270K data pool.
2. Apply LM-Mixup to each group of $n$ inputs to generate a smaller set of fused, high-quality candidates.
3. Compute long-tail diversity scores for all mixup-generated outputs.
4. Select the top 5K samples to form the final MIXUP high-quality dataset.

This process allows LM-Mixup to aggregate and refine weak signals into information-dense, high-quality supervision data.

## G   MORE EXPERIMENT RESULTS

**Full Low-Quality vs. Full Mixup Data.** To comprehensively evaluate the effectiveness of our approach, we conduct experiments on three representative models—Mistral-7B, LLaMA-3.1-8B, and LLaMA-2-7B-hf—using 10K samples drawn respectively from the raw low-quality dataset and the mixup-enhanced dataset generated via *LM-Mixup*. As shown in Figure 8, directly fine-tuning

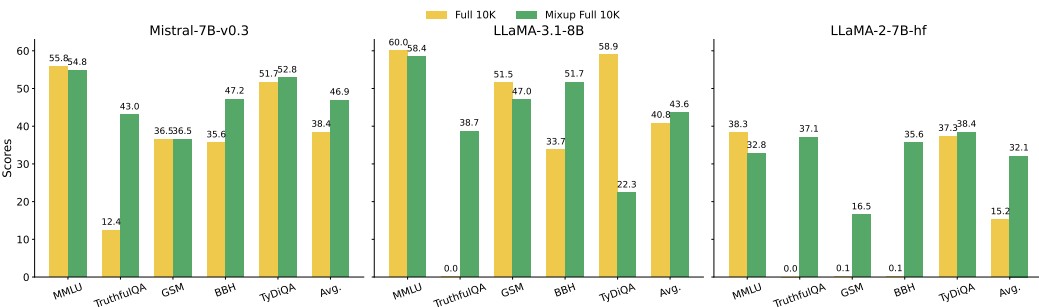

Figure 8: Comparison of model performance on five benchmarks using the full low-quality dataset versus the full mixup dataset (10K samples). Scores are reported for MMLU, TruthfulQA, GSM, BBH, TyDiQA, and the overall average.

on low-quality data leads to inconsistent and often suboptimal performance across most benchmarks. In contrast, the mixup-enhanced data substantially boosts performance on key tasks such as TruthfulQA, BBH, TyDiQA, and the overall average score for all three models. Notably, the improvements are most pronounced on LLaMA-2-7B-hf, where the baseline performance on raw data is particularly low, highlighting the robustness of *LM-Mixup* in challenging low-quality settings. These results collectively demonstrate that our method consistently transforms low-quality samples into a valuable resource for instruction tuning, unlocking their potential and significantly narrowing the gap with high-quality data baselines.

**Additional Results on LLaMA-2-7B-hf and Mistral-7B-v0.3.** We additionally conducted experiments to assess the performance of the OpenLLM leaderboard across different baseline settings using various backbone models, including Mistral-7B-v0.3 and LLaMA-2-7B-hf. Tables 5 and 6 report the corresponding results for these two backbones, respectively. Overall, the findings further confirm the effectiveness of our method, demonstrating that with appropriate configurations, it can consistently achieve top-2 performance on the leaderboard.

Table 5: Results on the OpenLLM leaderboard using `Mistral-7B-v0.3` as the base model. The top-performing scores are shown in **bold**, while the second-best scores are marked with underlines. * indicates that the values are sourced from Pang et al. (2024).

| Models | MMLU (factuality) | TruthfulQA (truthfulness) | GSM (reasoning) | BBH (reasoning) | TydiQA (multilinguality) | Average |
|---|---|---|---|---|---|---|
| VANILLA BASE MODEL* | 59.7 | 30.2 | 38.0 | 49.6 | 54.9 | 46.5 |
| COMPLETION LENGTH* | 58.9 | 34.4 | 42.5 | 53.1 | 59.6 | 49.7 |
| PERPLEXITY* | 59.8 | 40.3 | 36.0 | 48.9 | 57.4 | 48.5 |
| $k$-NN-10* | 58.3 | 41.7 | 43.5 | 54.1 | 53.4 | 50.2 |
| RANDOM SELECTION* | 59.4 | 36.7 | 41.8 | 54.2 | 54.0 | 49.3 |
| LESS* | 59.5 | 34.8 | 42.0 | 54.5 | 57.5 | 49.7 |
| FULL DATA (300K)* | 60.0 | 43.5 | 43.5 | 52.5 | 53.4 | 50.6 |
| ALPAGASUS* | 60.5 | 36.7 | 41.0 | 55.1 | 57.3 | 50.1 |
| DEITA* | 60.1 | 35.6 | 40.5 | 55.1 | 56.0 | 49.5 |
| DS2 W/O CURATION* | 60.1 | 35.9 | 48.5 | 54.2 | 58.9 | 51.5 |
| DS2* | 59.9 | 37.9 | 47.5 | 55.6 | 59.3 | **52.0** |
| MIXUP 70% + ORI 30% | 58.5 | 42.7 | 46.0 | 53.2 | 52.9 | 50.7 |
| MIXUP 50% + ORI 50% | 57.0 | 43.0 | 47.0 | 54.0 | 52.6 | 50.7 |
| MIXUP 30% + ORI 70% | 56.0 | 45.3 | 51.5 | 54.0 | 52.1 | 51.8 |

# H ADDITIONAL DATASET STATISTICS ABOUT MIXTURE

Figure 9 reports the distribution of the number of low-quality variants constructed for each high-quality sample. Most samples are paired with multiple degraded variants, enabling the model to learn hierarchical mappings from noisy or incomplete inputs to high-quality outputs.

Table 6: Results on the OpenLLM leaderboard using `LLaMA-2-7B-hf` as the base model. The top-performing scores are shown in **bold**, while the second-best scores are marked with underlines. * indicates that the values are sourced from Pang et al. (2024).

| Model | MMLU (factuality) | TruthfulQA (truthfulness) | GSM (reasoning) | BBH (reasoning) | TydiQA (multilinguality) | Average |
|---|---|---|---|---|---|---|
| VANILLA LLAMA-2-7B* | 41.9 | 28.4 | 6.0 | 38.3 | 35.7 | 30.1 |
| COMPLETION LENGTH* | 42.4 | 36.4 | 1.5 | 36.8 | 33.9 | 30.2 |
| PERPLEXITY* | 45.0 | 41.5 | 12.0 | 31.7 | 39.5 | 33.9 |
| $k$-NN-10* | 38.2 | 40.8 | 15.0 | 36.0 | 43.8 | 34.8 |
| RANDOM SELECTION* | 44.7 | 41.8 | 14.0 | 37.9 | 40.8 | 35.8 |
| LESS | 44.3 | 38.2 | 18.0 | 35.2 | 46.3 | 36.4 |
| FULL DATA (300K)* | 50.1 | 36.2 | 16.5 | 40.5 | 46.7 | 38.0 |
| ALPAGASUS* | 45.3 | 41.0 | 14.5 | 37.0 | 45.3 | 36.6 |
| DEITA* | 45.2 | 44.7 | 13.5 | 35.6 | 43.4 | 36.5 |
| DS2 W/O CURATION* | 42.0 | 39.5 | 15.0 | 38.1 | 46.1 | 36.1 |
| DS2* | 40.2 | 43.8 | 13.5 | 38.9 | 46.5 | 36.6 |
| MIXUP 70% + ORI 30% | 39.5 | 42.5 | 17.0 | 38.6 | 42.5 | 36.0 |
| MIXUP 50% + ORI 50% | 39.5 | 44.0 | 18.0 | 38 | 42.5 | 36.4 |
| MIXUP 30% + ORI 70% | 39.0 | 45.3 | 18.0 | 38.0 | 43.5 | **36.8** |

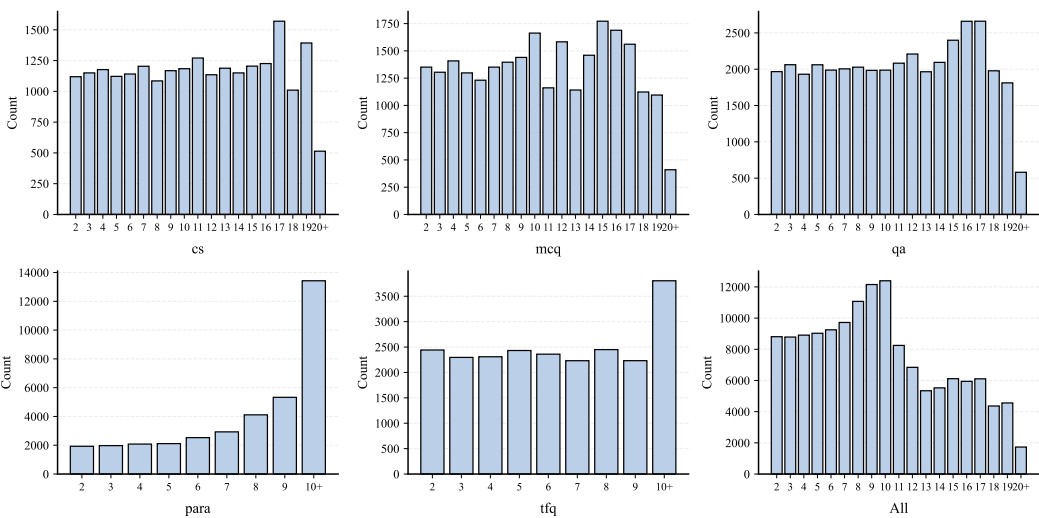

Figure 9: Distribution of the number of low-quality samples derived from each high-quality sample across different task categories.

Table 12 provides a detailed breakdown of the entire dataset across five task types (QA, MCQ, CS, TFQ, Paragraph) and three data variants (Normal, Cross-Topic, Noisy). We observe a balanced distribution across task types, with QA and Paragraph slightly larger in size, ensuring diverse coverage for training and evaluation.

# I   PROMPT TEMPLATE

The prompt template below illustrates how we use ChatGPT-4o-mini to generate high-quality data, with the following notes for different task types: For qa, let Instruction be the question and Output be the answer. For mcq, include options inside Instruction and provide the correct choice and a brief rationale in Output. For cs, tfq, or paragraph styles, keep Instruction as the task prompt and Output as the targeted response.

---

**Prompt Template for High Quality Data Generation**

You are a knowledgeable assistant tasked with producing **exceptionally high-quality {task_type}** instances that will later be rated on four axes: *Rarity*, *Complexity*, *Informativeness*, and *Overall*.

**Scoring Criteria**

- **Rarity (1–10):** Cover non-obvious, less-quoted aspects; avoid commonplace trivia.
- **Complexity (1–10):** Require synthesis of multiple facts, causal/temporal links, or non-trivial reasoning; avoid single-sentence lookups.
- **Informativeness (1–10):** Deliver dense, relevant, non-trivial content—even if concise; add value beyond superficial recall.
- **Overall (1–10):** Aggregate impression; aim for the top tier when justified.

**Requirements**

1. Generate **3–4** high-quality **{task_type}** instances based on the passage below.
2. For each instance, explicitly maximize the four criteria above (Rarity, Complexity, Informativeness, Overall).
3. You may quote or paraphrase the passage, but *weave* information to show reasoning and uniqueness; avoid verbatim copying when unnecessary.
4. Length is flexible—prioritize informativeness and reasoning over verbosity.
5. Use **plain text only** in the following exact format (no markdown):

```
Instruction: <instruction 1>
Output: <output 1>

Instruction: <instruction 2>
Output: <output 2>

Instruction: <instruction 3>
Output: <output 3>
```

**Passage**

```
{passage}
```

The prompt template below illustrates how we use ChatGPT-4o-mini to generate low quality data, with the following notes for different task types: For qa, let Instruction be the question and Output be the answer. For mcq, include options inside Instruction and provide the correct choice with minimal explanation in Output. For cs, tfq, or paragraph styles, keep Instruction as the task prompt and Output as the response, ensuring only moderate relevance and detail. The number n is determined by a random value.

**Prompt Template for Low Quality Data Generation**

You are a moderately skilled assistant tasked with producing **{n} low-quality {task_type}** instances derived from the given original data. Each generated instance should deliberately reflect *average quality*, aiming for scores between **3–6 (on a 10-point scale)** on four evaluation axes: *Rarity*, *Complexity*, *Informativeness*, and *Overall*.

**Scoring Criteria**

- **Rarity (3–6):** Cover reasonably common aspects; avoid both overly trivial and highly novel content.
- **Complexity (3–6):** Allow some light inference but avoid deep reasoning or multi-step logic.
- **Informativeness (3–6):** Ensure answers are mostly correct but lack nuance, depth, or precision.
- **Overall (3–6):** The overall impression should feel average, slightly useful, somewhat generic, and unpolished.

**Hard Constraints**

1. **Same Topic**: All {task_type} instances must stay on the identical topic as the original.
2. **Explicit Subject**: The main subject or event (e.g., *the execution of Turner and McDaniel*) *must be stated verbatim* in every question and answer. Avoid vague pronouns unless the noun is immediately repeated.
3. **Self-contained**: Each {task_type} must be understandable in isolation; assume no external context.
4. **No Off-topic Content**: Do not introduce unrelated domains or shift the factual focus.

**Output Requirements**

1. Generate **3–4** {task_type} instances based on the original data.
2. Maintain moderate quality (scores 3–6) on all four evaluation axes.
3. Use **plain text only** in the exact format below (no markdown):

```
Instruction: <instruction 1>
Output: <output 1>

Instruction: <instruction 2>
Output: <output 2>

Instruction: <instruction 3>
Output: <output 3>
```

**Original Data**

```
{orig}
```

The prompt template below illustrates how we use ChatGPT-4o-mini to perform data fusion across different task types, with the following notes: For qa, merge two question–answer pairs into a single, coherent question with a unified answer. For mcq, combine two multiple-choice questions into one integrated question, providing a single correct option with a concise explanation. For cs, tfq, or paragraph tasks, merge the content of both instances into a single prompt–response pair, ensuring the output reflects a natural synthesis of the original information while maintaining moderate length and relevance.

---

Prompt Template for Data Fusion across {task_type}

You are a data fusion expert tasked with merging two {task_type} instances into a single, coherent, and high-quality {task_type} instance. The goal is to synthesize both original samples into one unified output that naturally connects the information from both inputs while maintaining high quality.

**Fusion Requirements**

1. Merge the two original {task_type} samples into **one concise, integrated instance**.
2. The unified output must address both original inputs comprehensively while avoiding redundancy or contradiction.
3. The fusion should capture *subtle conceptual links*, rather than simply stacking facts together.
4. Ensure the final output meets the following quality criteria:
   - **Rarity:** Avoid overly common or trivial facts; focus on non-obvious insights.
   - **Complexity:** Encourage nuanced reasoning or implicit connections.
   - **Informativeness:** Maximize factual density and relevance.
   - **Overall Quality:** Aim for the top tier across all above dimensions.

**Output Format**

```
Instruction: <your merged instruction>
Output: <your merged output>
```

**Original Instances**

```
Instance-1:
{text1}

Instance-2:
{text2}
```

---

Following previous work (Pang et al., 2024), we use the same template for LLM Rating:

---

Prompt Template for LLM Rating

As a data quality estimator, your task is to assess the quality of the data sample based on the criteria: Rarity, Complexity, and Informativeness. Please rate the sample on a scale from 1 to 10 for each criterion, and return an overall rating on a scale from 1 to 10, where a higher score indicates a higher level of quality. Ensure that the ratings are not overly concentrated around a specific score. If multiple samples have similar qualities, consider spreading the scores more evenly to reflect subtle differences.

Please carefully evaluate the following data sample and return the integral evaluation scores using the JSON format:

```
{"Rarity": <number, 1-10>,
    "Complexity": <number, 1-10>,
    "Informativeness": <number, 1-10>,
    "Overall rating": <number, 1-10>}
```

Instruction: [Instruction]
Input: [Input]
Response: [Response]

---

## J  CASE STUDY

Tables 7, 8, 9, 10, and 11 present representative mixup cases across QA, CS, TFQ, MCQ, and paragraph tasks, illustrating how multiple fragmented inputs are integrated into coherent, high-quality outputs.

Table 7: QA pairs mixup example.

| Input (Multiple QA Pairs) |
| --- |
| Q: What are the financial implications Hardy discussed with the AAHA and Saskatchewan Amateur Hockey Association regarding the discussions on player development? |
| A: Hardy's discussions highlighted that financial implications include a strain on resources, showing the need for more investment in player retention to prevent losing talent to professional leagues. |
| Q: How might the discussions with the AAHA and Saskatchewan Amateur Hockey Association on player development affect the competitiveness of amateur hockey? |
| A: The discussions suggest that amateur hockey may face challenges to its competitiveness if it cannot adapt to the pressures from professional leagues, potentially leading to a decline in player quality. |
| Q: What strategic changes were suggested in Hardy's discussions with the AAHA and Saskatchewan Amateur Hockey Association regarding player development? |
| A: Hardy's discussions prompted the idea of re-evaluating investment strategies in player development to ensure that amateur hockey remains appealing and retains its talent pool amidst increasing professional competition. |
| Q: What broader challenges were mentioned by Hardy concerning the discussions on player development? |
| A: Hardy mentioned that broader challenges include maintaining the integrity of amateur hockey and ensuring grassroots involvement, which are at risk due to the rising influence of professional leagues. |

| Output (Merged Answer) |
| --- |
| Q: Analyze how Hardy's post-WWII discussions with amateur hockey associations reflected emerging tensions in Canadian sports culture, particularly regarding financial constraints, competitiveness, and societal identity shifts. |
| A: In examining Hardy's dialogue with prominent figures representing national governing bodies post-WWII, there emerges a profound intersectionality of personal ambition juxtaposed against systemic constraints faced by athletes transitioning towards professional careers. His candid remarks reveal underlying concerns about the sustainability of amateurism given escalating demands placed upon young talents competing internationally. Such sentiments reflect wider cultural shifts occurring globally wherein individuals sought greater recognition beyond localized boundaries—a trend mirrored in other sporting disciplines experiencing rapid modernization. Furthermore, Hardy's reflections hint at deeper socio-political ramifications tied to identity formation linked to sport participation, suggesting that individual aspirations intersected critically with collective narratives shaped by historical events affecting community cohesion. Overall, these discussions underscore significant transformations happening simultaneously in society and athletics, revealing complexities inherent in balancing tradition with contemporary realities facing aspiring amateurs navigating pathways toward professional success. |

Table 8: CS pairs mixup example.

| Input (Multiple CS Pairs) |
| --- |
| C: Historical evolution of electromagnetic theory
S: The historical evolution of electromagnetic theory involves key figures like Maxwell who explored the aether theory, which proposed that forces act through a medium.
C: Philosophical implications of electromagnetic theory
S: The philosophical implications of electromagnetic theory reflect a shift in understanding forces, moving from the aether concept to recognizing the nature of fields and potentials in physics.
C: Impact of electromagnetic theory on engineering
S: The impact of electromagnetic theory on engineering principles is substantial, as it provides a framework for understanding different physical properties compared to concepts like potential energy.
C: The role of experiments in electromagnetic theory
S: The role of experiments in electromagnetic theory, particularly the Michelson–Morley experiment, played a crucial part in dismissing the aether theory and reshaping modern physics. |
| **Output (Merged Answer)** |
| C: Synthesis of Physics Concepts
S: The exploration of electromagnetic phenomena reflects significant advancements in classical mechanics, wherein the transition from Newtonian dynamics towards relativistic formulations necessitated a paradigmatic change akin to shifting away from traditional force-based models toward field-theoretic approaches rooted in differential forms. While early attempts at explaining electrostatics involved complex interactions mediated via hypothetical entities known as electric fluids, later thinkers sought simpler explanations grounded in mathematical constructs involving vector calculus. Notably, the dismissal of the aether hypothesis postulated by Fresnel was pivotal, leading physicists to embrace alternative interpretations supported by extensive experimental validation, including notable outcomes observed in the Michelson–Morley interferometer experiment. These evolving understandings illustrate profound intersections between philosophy and science, revealing deeper layers of reality beyond conventional boundaries. |

Table 9: TFQ pairs mixup example.

| Input (Multiple TFQ Pairs) |
| --- |
| Joanne Peason's view of Pryce's essay does not show an important moment in the debate about Christianity's origins in Britain.
Answer: False
Joanne Peason believes that Pryce's essay fails to effectively challenge established legends.
Answer: False
Pryce's work is considered by Joanne Peason to have no relevance to discussions of Welsh identity and Christianity.
Answer: False |
| **Output (Merged Answer)** |
| Joanne Pearson's assertion that Pryce's essay represents "a significant point" in challenging prevailing myths suggests she recognizes the importance of questioning traditional accounts of early Christianization processes in Britain, especially relating to Welsh culture. Her belief indicates a shift towards more nuanced interpretations where local identities were intertwined with religious conversion stories during periods like Roman rule. Such skepticism contributes significantly to ongoing debates over national histories shaped by colonial influences, suggesting that Pryce's exploration may serve dual purposes: critiquing existing narratives and fostering contemporary dialogues about indigenous rights and heritage preservation amidst modernity.
Answer: False |

Table 10: MCQ pairs mixup exmaple.

| Input (Multiple MCQ Pairs) |
| --- |

Q: What is a possible effect of contestants giving "Bonus" points to each other throughout the show?
A) It might lead to an increased sense of competition among contestants.
B) Contestants may use Bonus points to create alliances, improving team dynamics.
C) The Bonus points could make contestants less motivated to help one another.
D) Contestants might ignore Bonus points and only care about the judges' scores.
Answer: B
Q: How do Bonus points awarded by contestants impact their relationships?
A) They may cause contestants to form rivalries that hurt their performance.
B) Contestants could leverage Bonus points to build alliances and boost collaboration.
C) The use of Bonus points may lower motivation among contestants.
D) Contestants generally overlook Bonus points, focusing only on judges' feedback.
Answer: B
Q: In the context of the show, how might the Bonus points affect contestant interactions?
A) They could lead to increased competition and less teamwork.
B) Contestants might use Bonus points to create positive alliances, enhancing performance.
C) The Bonus points system could demotivate contestants from giving constructive feedback.
D) Contestants may disregard Bonus points, concentrating on judges' assessments instead.
Answer: B
Q: What impact do Bonus points have on how contestants relate to one another?
A) They might foster rivalries that detract from overall performance.
B) Contestants may utilize Bonus points to form alliances, which can enhance their teamwork.
C) The Bonus points system could lead to a lack of motivation among the contestants.
D) Contestants often pay little attention to Bonus points, focusing more on judges' scores.
Answer: B

| Output (Merged MCQ Instance) |
| --- |

Q: Analyzing the psychological dimensions involved in the game structure reveals insights into how power operates differently than traditional authority figures. Consider the role of contestants awarding "Bonus" points to each other over time. Discuss the broader implications of this mechanism on perceptions of fairness and trustworthiness among competitors. Which statement best captures the dual nature of this dynamic?
A) Awarding Bonus points fosters genuine camaraderie among contestants, leading to improved performances due to mutual support rather than fear-based compliance typical of hierarchical setups.
B) By allowing contestants to give Bonus points freely, the hierarchy becomes irrelevant, suggesting that true leadership emerges solely from meritocratic principles regardless of initial status differences.
C) The ability to manipulate others through Bonus points creates distrust among teammates, undermining collaborative efforts necessary for success, thereby reinforcing existing hierarchies despite attempts to level playing fields.
D) Giving Bonus points reflects a shift towards egalitarianism, enabling weaker members to challenge dominant personalities directly, ultimately reshaping societal norms beyond just entertainment purposes.
Answer: C

Table 11: Paragraphs mixup example.

| **Input (Four Separate Paragraphs)** |
| --- |
| 1. The Industrial Revolution transformed urban landscapes across Europe, bringing rapid technological advancements but also leading to overcrowding and unsanitary living conditions in many cities. 2. During the same period, public health reforms emerged as governments recognized the need for clean water systems, waste management, and regulations to combat disease outbreaks. 3. Architects and city planners began integrating green spaces and wider streets into urban designs, seeking to improve both public health and aesthetic appeal. 4. These efforts laid the groundwork for modern urban planning, where functionality, sustainability, and livability became central considerations in shaping city environments. |
| **Output (Merged High-Quality Paragraph)** |
| The Industrial Revolution reshaped European cities through rapid industrialization, introducing new technologies but also creating overcrowded and unhealthy living conditions. As governments grappled with recurring disease outbreaks, public health reforms prioritized clean water systems, sanitation, and housing regulations. Urban planners responded by incorporating green spaces, broader streets, and organized infrastructure, aiming to balance industrial growth with livable environments. These combined efforts forged the foundations of modern urban planning, where health, sustainability, and functionality guide the evolution of contemporary cities. |

## K  DATASET STATISTICS

Table 12: Dataset statistics across five task categories. For each category, we report the total number of samples (train+test) and their breakdown into noisy, cross-Topic, and normal subsets. Ratios indicate the percentage of the full dataset.

| Category | Samples | Ratio (%) |
| --- | --- | --- |
| **I. cs** | **22,012** | **15.2** |
| • Noisy | 1,100 | 0.8 |
| • Cross-Topic | 4,852 | 3.3 |
| • Normal | 16,060 | 11.1 |
| **II. mcq** | **25,437** | **17.6** |
| • Noisy | 1,272 | 0.9 |
| • Cross-Topic | 4,714 | 3.3 |
| • Normal | 19,451 | 13.4 |
| **III. qa** | **38,455** | **26.6** |
| • Noisy | 1,922 | 1.3 |
| • Cross-Topic | 7,405 | 5.1 |
| • Normal | 29,128 | 20.1 |
| **IV. para** | **36,423** | **25.1** |
| • Noisy | 1,821 | 1.3 |
| • Cross-Topic | 8,261 | 5.7 |
| • Normal | 26,341 | 18.2 |
| **V. tfq** | **22,557** | **15.6** |
| • Noisy | 1,128 | 0.8 |
| • Cross-Topic | 4,154 | 2.9 |
| • Normal | 17,275 | 11.9 |
| **All** | **144,884** | **100.0** |

