# OpenReview forum: "LM-mixup: Text Data Augmentation via Language Model based Mixup"
_ICLR.cc/2026/Conference — Submitted to ICLR 2026_

### Official Review · Reviewer_s6k7 · 2025-10-27

**Soundness:** 2
**Presentation:** 1
**Contribution:** 2
**Rating:** 4
**Confidence:** 3

**Summary:**

This paper introduces Instruction Distillation, a new paradigm for improving the quality of low-quality instruction-following data. The authors propose a dataset called MIXTURE that maps multiple low-quality or redundant text inputs to a distilled high-quality target. Building on this dataset, they develop LM-Mixup, a reinforcement learning framework that fine-tunes language models using GRPO with three rewards. The method aims to transform low-quality, redundant, or noisy samples into information-dense outputs. Experimental results show that LM-Mixup outperforms SFT and several strong data selection baselines.

**Strengths:**

- The paper formalizes the Instruction Distillation task, clearly articulating the goal of transforming multiple imperfect inputs into a single high-quality instruction-output pair.

- Extensive experiments demonstrate consistent improvements across both in-domain (MIXTURE) and out-of-domain (OpenLLM leaderboard) benchmarks.

**Weaknesses:**

1. Limited Practical Applicability. The proposed setting assumes the availability of a large proportion of low-quality or redundant data, which may not reflect many real-world instruction-tuning scenarios where most data are already of moderate or acceptable quality. Consequently, the practical scope of Instruction Distillation could be narrower than implied, and its benefits may diminish when applied to more balanced or higher-quality datasets.

2. The paper’s exposition is at times vague and difficult to follow. For example, the details of the distillation process.


3. The experimental comparisons omit several intuitive and important baselines:

- Simple repair via LLM rewriting: directly improving low-quality samples using an LLM without mixup or GRPO.

- High-quality-only training: fine-tuning solely on the curated high-quality subset to measure the added value of the mixup process.

- Alternative data augmentation methods: such as paraphrasing or backtranslation.

Including these baselines would provide a clearer understanding of where LM-Mixup stands relative to simpler or more established augmentation techniques.

**Questions:**

See weaknesses

---

> ### Author Response · Authors · 2025-11-18
>
> Thank you for taking the time to review our work. We sincerely appreciate your constructive feedback, which has been invaluable in helping us identify areas for improvement. We are also grateful for your recognition of the strengths of our study. Below, we respond to the concerns raised and further clarify the key aspects of our contributions.
>
> >**Clarification on the Practical Applicability of Instruction Distillation (Response to W1):**
>
> We thank the reviewer for the comment. However, we would like to reiterate our motivation. In some standard instruction-tuning settings, the available data may indeed already be of acceptable quality. However, in real-world applications, **“high quality” is always a relative and scarce notion**. Several recent studies on high-quality data filtering, such as DS2 [1], LESS [2], and Alpagasus [3], consistently report that **only about 3% to 5% of samples in large-scale instruction-tuning corpora can be considered truly high quality**. Moreover, fine-tuning with only this top 3% to 5% already outperforms fine-tuning on the full dataset, and the remaining lower-quality data is typically discarded. At the same time, **in low-resource tasks, specialized domains such as law and medicine, and many long-tail instruction categories, the scarcity of high-quality data is even more pronounced**. In practice, the majority of real-world data often consists of “medium-to-low quality, noisy, and stylistically inconsistent” samples [4-6]. The motivation behind Instruction Distillation is precisely to address this structural mismatch by enabling models to systematically transform low-quality data into high-quality training signals without requiring additional human annotation.
>
> In addition, our MIXTURE dataset covers five types of downstream tasks (Section 3.2), and the experimental results in Section 5.1 further demonstrate that LM-mixup provides consistent improvements across these tasks. **These results indicate that the proposed method is not limited to specific scenarios and shows strong cross-task generalization ability.** For these reasons, we believe that the framework remains broadly applicable in real-world settings, especially in situations where high-quality data cannot be scaled up easily.
>
> > **Clarification of the Distillation Process (Response to W2):**
>
> We thank the reviewer for pointing out the lack of clarity in our description. This issue indeed arises from the current version not being sufficiently intuitive, and we will revise and expand the relevant sections in the updated manuscript. In brief, Instruction Distillation explicitly requires the model to aggregate information across multiple imperfect inputs and produce a semantically consistent and structurally valid high-quality output. Formally, we define the training set as $D = \\{ (X _ i, Y _ i) \\} _ {i=1}^n$, where each $X_i$ is a set of multiple low-quality inputs describing similar topics: $X _ i = \\{ x _ {i,1}, x _ {i,2}, \ldots, x _ {i,k _ i} \\}$. These inputs may be noisy, incomplete, or stylistically inconsistent, whereas $Y _ i$ is a single high-quality instruction-output pair. The generator $f_\theta$ is trained to map each input set to a refined output, $\hat{Y} _ i = f _ \theta(X _ i) = f _ \theta(\\{x _ {i,1}, x _ {i,2}, \ldots, x _ {i,k _ I}\\})$, thereby compressing several weak signals into one strong supervision signal, a formulation that is fundamentally different from data augmentation or curation. We will include more explicit textual explanations in the revised version to further improve clarity and readability.

---

> ### Author Response · Authors · 2025-11-18
>
> > **Additional Experiment Results (Response to W3):**
>
> We thank the reviewer for pointing out the experimental issues. We fully agree that these baselines are essential for a comprehensive evaluation of LM-Mixup. In response to your suggestions, we have incorporated additional experiments and also highlighted several baselines that were already included in the original submission. Below, we summarize the results point by point.
>
> ### **1. Simple LLM Rewriting Baselines**
>
> 1. In the original Table 1, we already included several untrained LLM baselines (e.g., Llama-3.1-8B-Instruct, DeepSeek-R1-Distill-Qwen-7B) to directly evaluate their capabilities on the MIXTURE test set. The results show that **LM-Mixup significantly outperforms these zero-shot models**. The detailed results are as follows:
>
>     | **Model** | **cs** | **mcq** | **para** | **qa** | **tfq** | **Avg** |
>     | --- | --- | --- | --- | --- | --- | --- |
>     | LLaMA-3.1-8B-Instruct | 3.61 | 2.57 | 3.57 | 3.71 | 2.10 | 3.27 |
>     | LLaMA-3.2-3B-Instruct | 3.58 | 2.66 | 3.52 | 3.78 | 2.49 | 3.21 |
>     | DeepSeek-R1-Distill-Qwen-7B | 3.61 | 2.46 | 3.40 | 3.23 | 2.41 | 3.02 |
>     | Qwen-2.5-7B-Instruct | 3.70 | 2.77 | 3.58 | 3.53 | 2.57 | 3.28 |
>     | Qwen-2.5-1.5B-Instruct | 3.39 | 2.44 | 3.34 | 3.33 | 1.34 | 2.86 |
>     | GPT-4o-mini | 3.81 | 2.86 | **3.69** | 3.64 | 2.61 | 3.37 |
>     | Qwen-2.5-1.5B-SFT | 3.54 | 3.25 | 2.82 | 3.73 | 3.05 | 3.28 |
>     | Qwen-2.5-7B-SFT | 3.53 | 3.31 | 3.41 | 3.78 | 3.10 | 3.46 |
>     | **LM-Mixup** | **3.85** | **3.55** | 3.31 | **4.17** | **3.32** | **3.66** |
> 2. To further explore the difference between “simple LLM rewriting” and LM-Mixup, we additionally compare LM-Mixup with OpenLLM-based rewriting. Specifically, we use the same base model as LM-Mixup (Qwen-1.5B-Instruct) to generate rewritten samples, replace the MIXUP portion in Table 3 with these rewritten samples, and conduct experiments under the exact same settings. All results are averaged over multi runs. **LM-Mixup consistently outperforms the simple rewriting strategy across all configurations,** confirming that the proposed method is substantially more effective. Results are as follows:
>
> | **Model** | **MMLU** | **TruthfulQA** | **GSM** | **BBH** | **TyDiQA** | **Avg** |
> | --- | --- | --- | --- | --- | --- | --- |
> | BASE 30% + ORI 70% | 61.3±0.7 | 38.2±0.7 | 51.2±1.4 | 54.2±0.9 | 23.0±0.5 | 45.6±0.4 |
> | MXIUP 30% +ORI 70% | 63.1±0.3 | 46.8±0.6 | 61.2±1.5 | 58.0±0.2 | 63.4±1.1 | **58.5±0.1** |
> | BASE 50% + ORI 50% | 62.1±0.1 | 37.4±0.7 | 50.8±0.6 | 54.1±0.7 | 22.9±0.4 | 45.4±0.2 |
> | MIXUP 50% + ORI 50% | 63.3±0.3 | 52.6±0.1 | 65.6±0.2 | 61.3±0.3 | 64.6±0.3 | **61.5±0.1** |
> | BASE 70 + ORI 70% | 62.2±0.9 | 40.7±0.5 | 54.3±0.2 | 55.1±1.1 | 23.2±0.3 | 47.1±0.1 |
> | MIXUP 70% + ORI 30% | 63.0±0.2 | 47.9±0.3 | 63.3±0.6 | 61.1±0.3 | 64.2±0.6 | **59.9±0.1** |
>
> ### **2. High-Quality-Only Training Baseline**
>
> **This comparison is actually already included in the original Table 3. The baselines comprise AlpaGasus [3], DEITA [7], and DS2 [1]**, all of which select the top-quality 10K subset from the 300K pool for training. Details are provided in Appendix D. Results are as follows:
>
> | **Method** | **MMLU** | **TruthfulQA** | **GSM** | **BBH** | **TydiQA** | **Avg** |
> | --- | --- | --- | --- | --- | --- | --- |
> | AlpaGasus | 63.4 | 42.6 | 66.0 | 59.1 | 59.4 | 58.1 |
> | DEITA | 64.5 | 50.1 | 60.0 | 60.3 | 63.7 | 59.7 |
> | DS2 w/o Curation | 63.3 | 51.5 | 62.0 | 59.7 | 64.3 | 60.2 |
> | DS2 | 64.0 | 50.3 | 67.5 | 59.0 | 66.1 | 61.4 |
> | MIXUP 70% + ORI 30% | 63.0±0.2 | 47.9±0.3 | 63.3±0.6 | 61.1±0.3 | 64.2±0.6 | 59.9±0.1 |
> | MIXUP 50% + ORI 50% | 63.3±0.3 | 52.6±0.1 | 65.6±0.2 | 61.3±0.3 | 64.6±0.3 | **61.5±0.1** |
> | MXIUP 30% +ORI 70% | 63.1±0.3 | 46.8±0.6 | 61.2±1.5 | 58.0±0.2 | 63.4±1.1 | 58.5±0.1 |

---

> > ### Author Response · Authors · 2025-11-18
> >
> > ### **3. Traditional Data Augmentation Baselines**
> >
> > We further include three commonly used augmentation baselines to more comprehensively validate the advantages of multi-source fusion: **(1) Back-translation [8], (2) LLM-based paraphrasing (using 4o-mini) [9], (3) EDA [10].** To ensure fair comparison, because traditional augmentation produces one-to-one rewrites, we sample 5K original samples from the pool and augment them to produce an additional 5K augmented samples, matching the total data size of the **50% MIXUP + 50% ORI** setting. All experiments are repeated three times.
> >
> > |**Method**  | **MMLU** | **TruthfulQA** | **GSM** | **BBH** | **TydiQA** | **Avg** |
> > | --- | --- | --- | --- | --- | --- | --- |
> > | Back-translation 50% + ORI 50% | 62.0±0.4 | 46.5±2.9 | 61.2±0.8 | 58.8±2.2 | 60.2±0.8 | 57.7±0.1 |
> > | EDA  50% + ORI 50% | 61.6±0.9 | 43.7±2.0 | 56.2±1.0 | 59.7±0.3 | 62.0±1.6 | 56.6±0.6 |
> > | Rephrasing 50% + ORI 50% | 61.4±0.7 | 36.0±2.5 | 63.2±1.0 | 59.6±0.2 | 62.2±0.8 | 56.5±0.6 |
> > | Mixup 70% + ORI 30% | 63.0±0.2 | 47.9±0.3 | 63.3±0.6 | 61.1±0.3 | 64.2±0.6 | 59.9±0.1 |
> > | Mixup 50% + ORI 50% | 63.3±0.3 | 52.6±0.1 | 65.6±0.2 | 61.3±0.3 | 64.6±0.3 | **61.5±0.1** |
> > | Mixup  30% + ORI 70% | 63.1±0.3 | 46.8±0.6 | 61.2±1.5 | 58.0±0.2 | 63.4±1.1 | 58.5±0.1 |
> >
> > These results show that **LM-Mixup consistently and significantly outperforms all augmentation baselines.** We believe the key reason is that traditional augmentation methods mainly perform surface-level or semantically equivalent rewrites, which do not introduce meaningful new information or improve semantic quality. In contrast, LM-Mixup exploits complementary information across multiple low-quality samples and performs semantic fusion that substantially increases information density and overall content quality, leading to superior downstream performance.

---

> ### Author Response · Authors · 2025-11-18
>
> References:
>
> [1] Pang J, Wei J, Shah A, et al. Improving Data Efficiency via Curating LLM-Driven Rating Systems[C]//The Thirteenth International Conference on Learning Representations.
>
> [2] Xia M, Malladi S, Gururangan S, et al. LESS: selecting influential data for targeted instruction tuning[C]//Proceedings of the 41st International Conference on Machine Learning. 2024: 54104-54132.
>
> [3] Chen L, Li S, Yan J, et al. AlpaGasus: Training a Better Alpaca with Fewer Data[C]//The Twelfth International Conference on Learning Representations.
>
> [4] Abdalla H B, Kumar Y, Marchena J, et al. The Future of Artificial Intelligence in the Face of Data Scarcity[J]. Computers, Materials & Continua, 2025, 84(1).
>
> [5] Dehouck M, Gómez-Rodríguez C. Data augmentation via subtree swapping for dependency parsing of low-resource languages[C]//28th international conference on computational linguistics. International Committee on Computational Linguistics; International Committee on Computational Linguistics, 2020: 3818-3830.
>
> [6] Alzubaidi L, Bai J, Al-Sabaawi A, et al. A survey on deep learning tools dealing with data scarcity: definitions, challenges, solutions, tips, and applications[J]. Journal of Big Data, 2023, 10(1): 46.
>
> [7] Liu W, Zeng W, He K, et al. What Makes Good Data for Alignment? A Comprehensive Study of Automatic Data Selection in Instruction Tuning[C]//The Twelfth International Conference on Learning Representations.
>
> [8] Edunov S, Ott M, Auli M, et al. Understanding Back-Translation at Scale[C]//Proceedings of the 2018 Conference on Empirical Methods in Natural Language Processing. 2018: 489-500.
>
> [9] Abaskohi A, Rothe S, Yaghoobzadeh Y. LM-CPPF: Paraphrasing-Guided Data Augmentation for Contrastive Prompt-Based Few-Shot Fine-Tuning[C]//Proceedings of the 61st Annual Meeting of the Association for Computational Linguistics (Volume 2: Short Papers). 2023: 670-681.
>
> [10] Wei J, Zou K. EDA: Easy Data Augmentation Techniques for Boosting Performance on Text Classification Tasks[C]//Proceedings of the 2019 Conference on Empirical Methods in Natural Language Processing and the 9th International Joint Conference on Natural Language Processing (EMNLP-IJCNLP). 2019: 6382-6388.

---

> ### Author Response · Authors · 2025-11-26
>
> Dear Reviewer s6k7,
>
> I hope this message finds you well.
>
> As the discussion phase is progressing, we would like to kindly check whether there are any remaining concerns or points that you would like us to further address. If there is anything else we can clarify or improve, please feel free to let us know. Your feedback has been extremely valuable, and we are happy to provide additional details to strengthen the paper.
>
> Thank you again for your time and constructive comments.
>
> Best regards,
>
> Authors

---

### Official Review · Reviewer_qBwT · 2025-10-27

**Soundness:** 2
**Presentation:** 3
**Contribution:** 3
**Rating:** 6
**Confidence:** 4

**Summary:**

The paper introduces a new task: instruction distillation, i.e., combining multiple low-quality instructions into a high-quality instruction. The authors then create a dataset for this task, where they trains a model with GRPO. They prove that the trained model is useful by applying it to improve the low-quality training data of other models. They observe an improvement on the performance when replacing the low-quality training data with distilled ones.

**Strengths:**

1. The task studied is meaningful and significant, and it would be great to have a high-quality dataset for the task
2. It's novel to train a model to do this specific task

**Weaknesses:**

1. I believe the task itself is not novel and there are papers, e.g., https://arxiv.org/pdf/2503.00034, that have discussed this problem. Yet these papers are not mentioned in the related works.
2. I do not fully understand the experiment setting. Please see question 1. This can potentially weaken the results of the experiments.
3. Although the authors have already explained it in section 5, LLM rating bias is still a valid concern. Specially, we see in the experiments that 50% low quality + 50% high quality data performs better than 30% low quality + 70% high quality. It's a sign that the rating system possibly has some issues. The authors' explanation is that "some specific portion of low quality data is good", but it doesn't make sense as "50% mixup + 50% high quality" improves over "50% low quality + 50% high quality". (mixup data should be of high quality by definition)

**Questions:**

1. LM-Mixup maps multiple low-quality examples into one high-quality example. So when you say things like "MIXUP 50% + ORI 50%", do you mean the 50% low-quality examples are condensed into fewer high-quality examples, or more low-quality examples are condensed into 50% high-quality examples, or somehow you transform 50% low-quality examples into 50% high-quality examples?
2. Can you give a proper explanation to the question raised in weakness 3, i.e., you said "Properly mixed low-quality data can even outperform high-quality-only baselines",  but then why can mixup (which should produce high-quality data) improve performance if that's the case?

---

> ### Author Response · Authors · 2025-11-18
>
> Thank you for taking the time to review our work. We sincerely appreciate your constructive feedback, which has been valuable in helping us identify areas for refinement. We are also grateful for your recognition of the task’s importance, the contribution of constructing a high-quality dataset, and the novelty of developing a model specifically designed for this problem. In the following, we respond to the concerns raised and clarify the key aspects of our contributions.
>
> > **Clarification on Novelty and Relation to Prior Work (Response to W1):**
>
> We thank the reviewer for pointing out this relevant work. We agree that at a high level there is a certain similarity between this paper and ours in the general idea of “using LLMs for data merging.” However, we would like to clarify that the two works differ fundamentally in task setup and methodological goals.
>
> First, MergeIT focuses on pairwise merging of existing medium- or high-quality instruction data, with the aim of increasing information density and compressing the overall data scale. Its method relies on semantic similarity and clustering structure to generate more concise and denser new instructions from pairs of semantically similar samples.
>
> In contrast, LM-Mixup tackles a fundamentally different problem: how to semantically fuse multiple thematically similar, noisy, and relatively low-quality answers into a single high-quality instruction. Unlike MergeIT’s pairwise setup, our input size is variable, resulting in a more general and challenging n → 1 fusion problem, as illustrated in Figure 9. This setting directly reflects the realistic scenario in which low-quality data is abundant while high-quality data is scarce. To address this challenge, we train a dedicated model using GRPO-based reinforcement learning with quality-, alignment-, and format-oriented rewards, enabling the model to explicitly improve information quality rather than simply compress semantically similar samples.
>
> In summary, although both methods involve the surface operation of “merging,” MergeIT addresses the compression of semantically similar, high-quality samples, whereas LM-Mixup tackles the quality improvement of multi-source low-quality samples. The two works differ in their input assumptions, task objectives, difficulty, and modeling mechanisms. We again thank the reviewer for the pointer and **will add a discussion of this paper in the Related Work section of the revised version**.
>
> > **Explanation on Experiment Setting (Response to W2, Q1)：**
>
> We thank the reviewer for raising this point. Due to space limitations, our original description of the experimental setup was not sufficiently detailed, which may have caused confusion,we apologize for that. Below, we provide a clearer explanation of how the data are constructed and what the proportions represent. We will also include this clarification in the appendix of the revised version to avoid ambiguity.
>
> We scored all 300K samples on a 1–5 quality scale. Among them, approximately 30K samples with scores ≥4 are treated as **high-quality data**, and roughly 270K samples with scores <4 are treated as **low-quality data**. In Table 3, the meaning of each part is as follows:
>
> 1. **ORI (High Quality Data)**:
>
>     From the 30K high-quality samples, we compute a long-tail score and select the top N as the ORI (high-quality-only) training set.
>
> 2. **LOW (Low Quality Data)**:
>
>     We randomly sample N instances from the 270K low-quality pool to form the LOW baseline.
>
> 3. **MIXUP (LM-Mixup Generated Data)**:
>
>     The MIXUP data are constructed as follows:
>
>     (1) Sample multiple groups of low-quality inputs from the 270K pool;
>
>     (2) Apply LM-Mixup to each group of *n* inputs to generate a smaller number of high-quality fused (mixup) samples;
>
>     (3) Compute long-tail scores for all mixup outputs;
>
>     (4) Select the top 5K as the MIXUP high-quality dataset.
>
>
> Therefore, the “MIXUP 50% + ORI 50%” setting means that the **MIXUP 50%** portion refers specifically to these 5K high-quality mixup samples generated via the above process.
>
> We will include a more explicit description of this data construction pipeline in the appendix of the final version to ensure the setup is clear and intuitive. Thank you again for your valuable feedback.

---

> ### Author Response · Authors · 2025-11-18
>
> > **Clarification on Our Experiments (Response to W3, Q2):**
>
> We thank the reviewer for carefully examining the details of our results. We apologize for the inappropriate wording in the original paper, which led to a misunderstanding, and we will revise it in the final version. What we intended to express is that the **low-quality data, after being fused by LM-mixup, is transformed into high-quality distillation signals, which means that the mixup data itself is good**. We **did not mean to suggest that a subset of the raw low-quality data is inherently good.** We will clarify the wording more precisely.
>
> Regarding the experimental setup, we noted in the paper that all 300K samples were assigned a quality score from 1 to 5. Approximately 30K samples with a score of 4 or above are considered high-quality, and the remaining roughly 270K samples are considered low-quality. In the LOW + ORI (HIGH) configurations, the “low-quality” portion is obtained by random sampling from the 270K low-quality pool, while the “high-quality” portion is taken from the 30K high-quality data by selecting the top-K samples according to long-tail scores. Because the low-quality pool is very large and each ratio configuration uses independent random sampling, it is possible that in some runs the 50% LOW + 50% ORI setting performs slightly better than the 30% LOW + 70% ORI setting. **This variation is due to statistical sampling variance and should not be interpreted as evidence that the scoring mechanism is systematically failing. To reduce such variance and obtain more stable conclusions, we repeated all experiments with multiple random seeds and reported the mean and standard deviation.**
>
> | **Main Table** | **MMLU** | **TruthfulQA** | **GSM** | **BBH** | **TydiQA** | **Avg** |
> | --- | --- | --- | --- | --- | --- | --- |
> | LOW 70% + ORI 30% | 62.7 ± 0.7 | 17.8 ± 2.0 | 62.5 ± 4.0 | 60.3 ± 0.9 | 65.6 ± 1.1 | 53.6 ± 1.2 |
> | MIXUP 70% + ORI 30% | 63.0 ± 0.2 | 47.9 ± 0.3 | 63.3 ± 0.6 | 61.1 ± 0.3 | 64.2 ± 0.6 | 59.9 ± 0.1 |
> | LOW 50% + ORI 50% | 62.4 ± 0.6 | 39.0 ± 9.8 | 62.7 ± 1.1 | 61.0 ± 2.2 | 64.0 ± 0.3 | 57.9 ± 2.0 |
> | MIXUP 50% + ORI 50% | 63.3 ± 0.3 | 52.6 ± 0.1 | 65.6 ± 0.2 | 61.3 ± 0.3 | 64.6 ± 0.3 | **61.5 ± 0.1** |
> | LOW 30% + ORI 70% | 60.9 ± 2.1 | 41.1 ± 5.6 | 62.7 ± 1.9 | 59.9 ± 1.7 | 60.4 ± 1.7 | 57.0 ± 0.8 |
> | MIXUP 30% + ORI 70% | 63.1 ± 0.3 | 46.8 ± 0.6 | 61.2 ± 1.5 | 58.0 ± 0.2 | 63.4 ± 1.1 | 58.5 ± 0.1 |
>
> From the averaged results, **the differences among these settings fall within a reasonable variance range, and the trends remain consistent with our overall conclusions.** In the final version, we will further clarify the sampling process and variance analysis to make the experiments and explanations more transparent.

---

> ### Author Response · Authors · 2025-11-26
>
> Dear Reviewer qBwT,
>
> I hope this message finds you well.
>
> As we approach the end of the discussion window, we wanted to make sure we’ve fully responded to your comments. If there is anything else you would like us to elaborate on, we would greatly appreciate the chance to clarify. Your insights have been tremendously helpful for improving our submission.
>
> Thank you for your thoughtful review.
>
> Best regards,
>
> Authors

---

> ### Comment · Reviewer_qBwT · 2025-11-26
>
> Thanks for your clarification. My question 2 has been properly addressed. For question 1, I think I understand your setting now. If I understand it correctly, you are missing 2 trivial baselines: one is 100% ORI, where you randomly sample 10K high quality data from the 30K; another one is the 10K data with the highest score, as you are selecting the mixup data with the highest score, instead of randomly sampling mixup data with score >=4.

---

> ### Author Response · Authors · 2025-11-27
>
> Thank you for your valuable comment. If we understand your concerns correctly, you mainly focus on two important baselines that we have not fully covered. Based on this, we respond from the following two aspects and supplement the experimental results on LLaMA-3.1-8B.
>
> **1) 100% Random Selected ORI (High Quality Data) Baseline**
>
> If we understand correctly, your first concern is that, in addition to the existing ORI selection methods based on highest-quality sorting, we should also include a baseline that **randomly selects 10K ORI samples from the high-quality data pool**.
>
> We have added this baseline accordingly. In addition, we would like to point out that the *AlpaGasus* [1], *DEITA* [2], and *DS2* [3] methods already included in our original paper are essentially methods that select the **top-10K highest-scoring samples** from high-quality ORI data, which are similar to the “highest-score ORI baseline” you suggested.
>
> The experimental results are as follows:
>
> | Method              | MMLU     | TruthfulQA | GSM      | BBH      | TyDiQA   | Avg          |
> | ------------------- | -------- | ---------- | -------- | -------- | -------- | ------------ |
> | Random high quality | 56.4     | 40.0       | 53.5     | 51.0     | 19.4     | 44.1         |
> | AlpaGasus           | 63.4     | 42.6       | 66.0     | 59.1     | 59.4     | 58.1         |
> | DEITA               | 64.5     | 50.1       | 60.0     | 60.3     | 63.7     | 59.7         |
> | DS2 w/o Curation    | 63.3     | 51.5       | 62.0     | 59.7     | 64.3     | 60.2         |
> | DS2                 | 64.0     | 50.3       | 67.5     | 59.0     | 66.1     | 61.4         |
> | MIXUP 70% + ORI 30% | 63.0±0.2 | 47.9±0.3   | 63.3±0.6 | 61.1±0.3 | 64.2±0.6 | 59.9±0.1     |
> | MIXUP 50% + ORI 50% | 63.3±0.3 | 52.6±0.1   | 65.6±0.2 | 61.3±0.3 | 64.6±0.3 | **61.5±0.1** |
> | MIXUP 30% + ORI 70% | 63.1±0.3 | 46.8±0.6   | 61.2±1.5 | 58.0±0.2 | 63.4±1.1 | 58.5±0.1     |
>
> It can be seen that our method is comparable to or exceeds the baselines that rely solely on selecting high-quality ORI data.
>
> **2) Baseline of Using Only 10K Mixup Data**
>
> If we understand correctly, your second concern is that we should include a baseline that **trains using only 10K mixup data**.
>
> We have already included this experiment in Appendix E, and we further present the results here (Figure 8 in the paper also visualizes these results). Since mixup data is generated from low-quality data, we use 10K low-quality data as the comparison baseline for fairness.
>
> **Mistral-7B-v0.3**
>
> | Method         | MMLU | TruthfulQA | GSM  | BBH  | TyDiQA | Avg      |
> | -------------- | ---- | ---------- | ---- | ---- | ------ | -------- |
> | Full Low 10K   | 55.8 | 12.4       | 36.5 | 35.6 | 51.7   | 38.4     |
> | Full Mixup 10K | 54.8 | 43.0       | 36.5 | 47.2 | 52.8   | **46.9** |
>
> **LLaMA-3.1-8B**
>
> | Method         | MMLU | TruthfulQA | GSM  | BBH  | TyDiQA | Avg      |
> | -------------- | ---- | ---------- | ---- | ---- | ------ | -------- |
> | Full Low 10K   | 60.0 | 0.0        | 51.5 | 33.7 | 58.9   | 40.8     |
> | Full Mixup 10K | 58.4 | 38.7       | 47.0 | 51.7 | 22.3   | **43.6** |
>
> **LLaMA-2-7B-hf**
>
> | Method         | MMLU | TruthfulQA | GSM  | BBH  | TyDiQA | Avg      |
> | -------------- | ---- | ---------- | ---- | ---- | ------ | -------- |
> | Full Low 10K   | 38.3 | 0.0        | 16.5 | 35.6 | 37.3   | 15.2     |
> | Full Mixup 10K | 32.8 | 37.1       | 0.1  | 38.4 | 38.4   | **32.1** |
>
> It can be clearly seen from the results that training solely on mixup data already significantly outperforms directly using low-quality data, which further supports our conclusion that mixup can effectively improve the utility of low-quality data.
>
> We again thank you for your insightful comment. If we still have any misunderstanding of your concerns, please feel free to let us know, and we will promptly supplement or clarify the results.
>
> **References:**
>
> [1] Chen L, Li S, Yan J, et al. AlpaGasus: Training a Better Alpaca with Fewer Data[C]//The Twelfth International Conference on Learning Representations.
>
> [2] Liu W, Zeng W, He K, et al. What Makes Good Data for Alignment? A Comprehensive Study of Automatic Data Selection in Instruction Tuning[C]//The Twelfth International Conference on Learning Representations.
>
> [3] Pang J, Wei J, Shah A, et al. Improving Data Efficiency via Curating LLM-Driven Rating Systems[C]//The Thirteenth International Conference on Learning Representations.

---

### Official Review · Reviewer_5cxv · 2025-10-31

**Soundness:** 3
**Presentation:** 3
**Contribution:** 2
**Rating:** 2
**Confidence:** 4

**Summary:**

This paper introduces LM-mixup, a method for augmenting low-quality instruction data by distilling multiple imperfect inputs into high-quality outputs using a language model fine-tuned with reinforcement learning. The authors construct Mixture, a 144K-sample dataset, and train LM-mixup using GRPO with multi-dimensional rewards. Experiments show that training on a small mixup-augmented subset (∼3% of full data) can match or exceed full-dataset training and compete with data selection baselines on OpenLLM benchmarks.

**Strengths:**

Practical Value: Demonstrates that low-quality data can be effectively repurposed to reduce dataset size and cost.

Comprehensive Experiments: Ablation studies and multiple benchmark evaluations strengthen empirical claims.

Reproducibility: Detailed training pipelines and dataset construction steps are provided.

**Weaknesses:**

Limited Academic Depth: The method feels like a well-engineered application of existing tools rather than a theoretical or algorithmic advance.

Weak Motivation: The paper does not sufficiently articulate why this particular form of "instruction distillation" is needed or how it differs philosophically from prior data augmentation or curation work.

Presentation Flaws: Inconsistent referencing and occasionally overly technical writing reduce readability and scholarly rigor.

**Questions:**

Could you better motivate the instruction distillation task in terms of its conceptual novelty beyond existing data augmentation or curation paradigms?

Have you considered comparing with simpler augmentation strategies (e.g., back-translation, paraphrasing) to better isolate the benefit of your multi-source fusion approach?

The reference formatting is inconsistent. Could you revise it to meet standard citation conventions?

How does LM-mixup ensure that distilled samples do not simply memorize or overfit to the high-quality targets, especially given the use of RL with reference-based rewards?

---

> ### Author Response · Authors · 2025-11-18
> **Reply to Reviewer 5cxv**
>
> Thank you for the time and effort in reviewing our work. We sincerely appreciate your constructive feedback, which has helped us identify areas for improvement. We are also grateful for your recognition of the practical value of repurposing low-quality data, the comprehensiveness of our experimental evaluations, and the reproducibility enabled by our detailed pipeline description. Below, we address the concerns raised and clarify key aspects of our contributions.
>
> > **Clarification of Academic Depth and Novel Contributions (Response to W1)**
>
> Thank you for your thoughtful comments. We would like to take this opportunity to restate the core motivation of our work. In recent years, instruction-tuning research has shifted from “using more data” to “fine-tuning with only a small amount of high-quality data.” However, **high-quality data remain scarce and expensive to obtain, while low-quality data constitute the overwhelming majority of real-world corpora.** Moreover, in many specialized or low-resource settings (e.g., low-resource machine translation, and domain-specific tasks such as law or medicine), **the scarcity of high-quality data is widely recognized as a key bottleneck limiting progress** [1-3]. Our work aims to address this fundamental pain point: by transforming abundant low-quality data into high-quality supervision through data fusion, we hope to unlock broader advances across multiple domains.
>
> Our work is consistent with many recent efforts such as SWE-GPT [4], ClarifyGPT [5], and Med-PaLM [6], which train or adapt large language models to become strong solvers for specific domains. Similar to these works, LM-Mixup also builds a task-oriented training pipeline. However, our focus differs in that we operate at the data layer: we design a pipeline that distills high-quality supervision from abundant low-quality corpora through multi-input fusion and quality-aware optimization, directly addressing the fundamental bottleneck of scarce high-quality instruction data.
>
> Our work also contributes two key technical innovations. First, we construct MIXTURE, a large-scale dataset that provides hierarchical mappings from multiple low-quality variants to their corresponding high-quality outputs. This design captures realistic imperfections in instruction data and enables supervised learning of many-to-one information fusion, an ability that existing instruction-tuning corpora do not support. Second, at the algorithmic level, we develop a multi-dimensional reward scheme tailored for instruction distillation, including a KNN–Bayes calibrated quality reward, a semantic alignment reward, and a format consistency reward. This reward design stabilizes RL training, mitigates noisy rating signals, and explicitly encourages quality improvement, which is an aspect rarely investigated in prior data-centric LLM research.

---

> ### Author Response · Authors · 2025-11-18
> **Reply to Reviewer 5cxv**
>
> > **Clarification on Our Motivation and the Distinction Between Instruction Distillation and Data Augmentation/Curation (Response to W2, Q1)**
>
> We sincerely apologize for the lack of clarity in our original writing, which may have led to a misunderstanding of our contribution. As we clarified in W1, **the goal of our proposed instruction distillation is to address the pervasive scarcity of high-quality data in many NLP scenarios, particularly in low-resource and domain-specific tasks.** By enabling the transformation of abundant low-quality data into high-quality supervision, our method aims to support the broader development of the low-resource NLP community.
>
> In addition, our MIXTURE dataset defines five distinct task types (see Sec. 3.2), and the experimental results in Sec. 5.1 demonstrate that LM-mixup achieves consistently strong performance across heterogeneous tasks. **These results highlight the generality and robustness of our proposed approach, beyond a single-task or purely engineering-oriented setting.**
>
> Importantly, our instruction distillation paradigm is fundamentally different from traditional data augmentation or data curation. For clarity, we provide the conceptual distinctions below:
>
> - **Instruction Distillation:** Extracts useful information from multiple low-quality or inconsistent responses, and generates high-quality instruction data through semantic fusion and quality-control mechanisms, thereby providing stronger learning signals for downstream models.
> - **Data Augmentation:** Generates additional samples via transformations such as rewriting, paraphrasing, cross-lingual translation, or synthetic sampling, with the primary goal of expanding data volume and enhancing diversity and generalization [7~8].
> - **Data Curation:** Improves or corrects labels for existing samples without modifying their semantic content, typically to enhance annotation quality or consistency [9~10].
>
> Instruction distillation fundamentally differs from data augmentation and data curation in that **it reduces data volume through information aggregation, changes and enriches the information content via semantic fusion, and explicitly fuses multiple low-quality sources into a single high-quality supervision signal.** As a result, unlike augmentation, which merely increases diversity, or curation, which only improves labels, instruction distillation directly improves content quality and delivers substantial gains in low-resource settings by converting many weak signals into few high-information-density instructions.
>
> We will also include additional details in the revised version to make the presentation clearer and more intuitive. We sincerely appreciate the reviewer’s insightful comment.
>
> > **Further Comparison with Data Augmentation Baselines (Response to Q2)**
>
> We thank the reviewer for this valuable suggestion. In response, we have added three widely used data augmentation baselines to more thoroughly evaluate the advantages of our multi-source fusion approach: **(1) Back-translation [11], (2) LLM-based paraphrasing (using *4o-mini*) [12], and (3) EDA [13].**
>
> To ensure a fair comparison, since conventional augmentation methods operate in a one-to-one rewriting or transformation manner, we first randomly sample **5K original instances** from the data pool and apply each augmentation method to generate an additional **5K augmented samples**. All experiments were repeated multiple times, and we report mean and standard deviation. The results are shown below:
>
> | **Method** | **MMLU** | **TruthfulQA** | **GSM** | **BBH** | **TydiQA** | **Avg** |
> | --- | --- | --- | --- | --- | --- | --- |
> | Back-translation 50% + ORI 50% | 62.0±0.4 | 46.5±2.9 | 61.2±0.8 | 58.8±2.2 | 60.2±0.8 | 57.7±0.1 |
> | EDA  50% + ORI 50% | 61.6±0.9 | 43.7±2.0 | 56.2±1.0 | 59.7±0.3 | 62.0±1.6 | 56.6±0.6 |
> | Rephrasing 50% + ORI 50% | 61.4±0.7 | 36.0±2.5 | 63.2±1.0 | 59.6±0.2 | 62.2±0.8 | 56.5±0.6 |
> | Mixup 50% + ORI 50% | 63.3±0.3 | 52.6±0.1 | 65.6±0.2 | 61.3±0.3 | 64.6±0.3 | **61.5±0.1** |
>
> These results show that **LM-Mixup consistently and significantly outperforms all augmentation baselines.** We believe the key reason is that traditional augmentation methods mainly perform surface-level or semantically equivalent rewrites, which do not introduce meaningful new information or improve semantic quality. In contrast, LM-Mixup exploits complementary information across multiple low-quality samples and performs semantic fusion that substantially increases information density and overall content quality, leading to superior downstream performance.

---

> ### Author Response · Authors · 2025-11-18
>
> > **Clarification on Overfitting and Memorization Concerns (Response to Q4):**
>
> Thank you very much for raising this important point. When designing LM-Mixup, **we indeed carefully considered the risks of overfitting and memorization, and we implemented several mechanisms to systematically prevent the model from relying on any single high-quality reference answer:**
>
> 1. **Dataset Diversification:**
>
>     During the construction of MIXTURE, we deliberately introduce cross-topic sample mixing and surface-level noise perturbations. **This increases stylistic and semantic diversity in the training distribution**, reducing the likelihood that the model overfits to fixed templates or specific target answers.
>
> 2. **Inherent Properties of the LM-Mixup Mechanism:**
>
>     In LM-Mixup, the model never learns from a *single* instruction–answer pair. Instead, each instance contains multiple low-quality responses, and there exist *many* valid high-quality fusion outputs for the same multi-source input. **This makes the objective naturally “one-to-many,” inherently weakening reliance on any specific reference answer.**
>
>     In addition, as noted in lines 87-89, GRPO differs fundamentally from SFT—which is more prone to “copy-the-answer” behavior. SFT performs token-level regression toward a single reference output, whereas GRPO generates multiple candidate responses per input, ranks them, and optimizes the policy from *relative preference signals*. **This mechanism reduces memorization of any one reference answer and instead encourages learning more generalizable fusion behaviors.**
>
> 3. **Similarity-based Reward Constraint to Prevent Reward Hacking:**
>
>     As shown in Section 5.3 (“Ablation on Reward Components”), using only a quality-rating reward can indeed cause the model to imitate high-quality answers too closely. To address this, we introduce an explicit **similarity-based reward constraint** to penalize excessive deviation from the original multi-source inputs. Our ablation results demonstrate that this similarity term is crucial for stabilizing training and suppressing overfitting.
>
> 4. **Generalization Evidence on OOD and Unseen Data:**
>
>     On the held-out portion of MIXTURE, LM-Mixup maintains strong performance **even on completely unseen samples**. Furthermore, our evaluations on **OpenLLM Benchmark OOD settings** show no signs of memorization; instead, the model learns broadly applicable patterns for generating high-quality outputs.
>
> > **Response to presentation concerns (Response to W3, Q3):**
>
> Thank you for the reviewer’s careful suggestions regarding the clarity of our presentation. Concerning the seemingly inconsistent referencing styles, we would like to clarify that this distinction was intentional and based on syntactic considerations: **when a citation functions as the grammatical subject or core semantic unit of the sentence, we use \citep to maintain natural sentence flow; in other contexts, we use \cite to preserve readability and rhythm.** This distinction was made to improve textual clarity rather than being an oversight. Nonetheless, we fully understand the reviewer’s concern about uniformity, and we will harmonize and refine the citation formatting in the revised version to ensure a more consistent and professional presentation.
>
> Regarding the comment on “*occasionally overly technical writing”*, we greatly appreciate the feedback. We will conduct a thorough review of the manuscript to simplify overly technical or overly long passages and further enhance readability. If the reviewer could kindly point out any specific sections of concern, we would be more than grateful and would refine them with greater precision.

---

> ### Author Response · Authors · 2025-11-18
>
> References:
>
> [1] Abdalla H B, Kumar Y, Marchena J, et al. The Future of Artificial Intelligence in the Face of Data Scarcity[J]. Computers, Materials & Continua, 2025, 84(1).
>
> [2] Dehouck M, Gómez-Rodríguez C. Data augmentation via subtree swapping for dependency parsing of low-resource languages[C]//28th international conference on computational linguistics. International Committee on Computational Linguistics; International Committee on Computational Linguistics, 2020: 3818-3830.
>
> [3] Alzubaidi L, Bai J, Al-Sabaawi A, et al. A survey on deep learning tools dealing with data scarcity: definitions, challenges, solutions, tips, and applications[J]. Journal of Big Data, 2023, 10(1): 46.
>
> [4] Ma Y, Cao R, Cao Y, et al. SWE-GPT: A Process-Centric Language Model for Automated Software Improvement[J]. Proceedings of the ACM on Software Engineering, 2025, 2(ISSTA): 2362-2383.
>
> [5] Mu F, Shi L, Wang S, et al. Clarifygpt: A framework for enhancing llm-based code generation via requirements clarification[J]. Proceedings of the ACM on Software Engineering, 2024, 1(FSE): 2332-2354.
>
> [6] Singhal K, Azizi S, Tu T, et al. Large language models encode clinical knowledge[J]. Nature, 2023, 620(7972): 172-180.
>
> [7] Feng S Y, Gangal V, Wei J, et al. A Survey of Data Augmentation Approaches for NLP[C]//Findings of the Association for Computational Linguistics: ACL-IJCNLP 2021. 2021: 968-988.
>
> [8] Chen J, Tam D, Raffel C, et al. An empirical survey of data augmentation for limited data learning in nlp[J]. Transactions of the Association for Computational Linguistics, 2023, 11: 191-211.
>
> [9] Northcutt C G, Athalye A, Mueller J. Pervasive Label Errors in Test Sets Destabilize Machine Learning Benchmarks[C]//Thirty-fifth Conference on Neural Information Processing Systems Datasets and Benchmarks Track (Round 1).
>
> [10] Karimi D, Dou H, Warfield S K, et al. Deep learning with noisy labels: Exploring techniques and remedies in medical image analysis[J]. Medical image analysis, 2020, 65: 101759.
>
> [11] Edunov S, Ott M, Auli M, et al. Understanding Back-Translation at Scale[C]//Proceedings of the 2018 Conference on Empirical Methods in Natural Language Processing. 2018: 489-500.
>
> [12] Abaskohi A, Rothe S, Yaghoobzadeh Y. LM-CPPF: Paraphrasing-Guided Data Augmentation for Contrastive Prompt-Based Few-Shot Fine-Tuning[C]//Proceedings of the 61st Annual Meeting of the Association for Computational Linguistics (Volume 2: Short Papers). 2023: 670-681.
>
> [13] Wei J, Zou K. EDA: Easy Data Augmentation Techniques for Boosting Performance on Text Classification Tasks[C]//Proceedings of the 2019 Conference on Empirical Methods in Natural Language Processing and the 9th International Joint Conference on Natural Language Processing (EMNLP-IJCNLP). 2019: 6382-6388.

---

> ### Author Response · Authors · 2025-11-26
>
> Dear Reviewer 5cxv,
>
> I hope this message finds you well.
>
> As the discussion period is ongoing and time is running short, we wanted to ensure we have addressed all your concerns satisfactorily. If there are any additional points or feedback you'd like us to consider, please let us know. Your insights are invaluable to us, and we're eager to address any remaining issues to improve our work.
>
> Thank you for your time and effort in reviewing our paper.
>
> Best regards,
>
> Authors

---

### Author Response · Authors · 2025-11-29
**General Response to the Area Chair and Reviewers**

Dear ICLR PC, SAC, AC and all reviewers,

We are deeply grateful for the efforts of the chairs and for the reviewers’ thoughtful and constructive feedback. We greatly appreciate your recognition of the importance of reusing low-quality instruction data, the contribution of constructing a new dataset tailored to this problem, and the empirical strength and reproducibility of our pipeline. Your comments have helped us substantially improve the clarity, positioning, and completeness of this work.

Across the reviews, the main concerns focus on the following aspects:
(1) the motivation and conceptual novelty of our proposed instruction distillation paradigm relative to existing data augmentation and data selection methods;
(2) the clarity of data construction and experimental configurations (e.g., the precise meaning of “MIXUP 50% + ORI 50%”);
(3) the potential bias of LLM-based rating and the interpretation of different low-/high-quality mixture ratios; and
(4) the need for comparisons with simpler or more established baselines, such as direct LLM rewriting, high-quality-only training, and traditional data augmentation methods.

In the revised version, we have made the following improvements:
(1) further clarified the practical applicability and motivation of our method (see Section 1);
(2) clearly distinguished instruction distillation from data augmentation and data curation, and elaborated the conceptual differences (see Appendix B);
(3) substantially expanded the description of our data scoring, sampling, and mixing pipeline, and moved the full protocol to the appendix for reproducibility (see Appendix E and F);
(4) added new baseline experiments, including back-translation, 4o-mini paraphrasing, EDA, and simple LLM rewriting using the same base model (see Section 5.2);
(5) provided a more detailed analysis of rating variance and mixture-ratio behavior, together with additional experimental results and clearer explanations to avoid misinterpretation (see Appendix F);
(6) revised several passages to improve clarity and avoid ambiguous or overly technical expressions (e.g., Section 3.1), and further expanded the discussion of related work.

We hope these clarifications and additions adequately address the reviewers’ concerns, and we sincerely appreciate your efforts in helping us improve the paper.

Best regards,

Authors

---

### Meta-Review · Area_Chair_wDkY · 2026-01-06

**Summary:**

The submission introduces LM-mixup, a data augmentation framework for Instruction Distillation designed to refine low-quality instruction data into high-quality signals via the MIXTURE dataset and GRPO. While the reviewers recognize the dataset's practical value and the framework's empirical effectiveness, critical concerns persist regarding the work's conceptual depth, the necessity of such a complex distillation paradigm compared to more direct curation methods, and potential rating biases inherent in the evaluation. Ultimately, while the work is technically well-executed, the consensus is that the submission requires further revision and a more distinct methodological contribution to fully meet the criteria of the conference.

**Reviewer Concerns:**

Addressed:
- The authors actively addressed experimental gaps by incorporating several competitive baselines and provided statistical mean and variance to demonstrate results' stability.
- They also clarified the conceptual definition of "Instruction Distillation" to distinguish it from traditional augmentation and curation.

Outstanding:
- The work is viewed as a well-engineered pipeline with limited novelty. Reviewers maintained that applying existing tools to this task is a mature application rather than a core innovation.
- The practical necessity of this distillation paradigm remains insufficiently justified compared to simpler strategies like data selection.
- Concerns persist regarding the reliance on LLM-based rewards and metrics, which may lead to stylistic "reward hacking" rather than objective gains.

**Reviewer Scores:**

Reviewer qBwT: Recognized the task's novelty. While the reviewer identified missing baselines in a late comment, the authors subsequently provided these results, further strengthening the empirical claims.

Reviewer s6k7: Remained skeptical about the practical applicability and the marginal value over established data-selection techniques.

Reviewer 5cxv: Maintained a high-confidence reject, identifying a lack of conceptual novelty and characterizing the method as a practical application of existing tools.

---

### Decision · Program_Chairs · 2026-01-26

Reject